# The Fair Value of Data Under Heterogeneous Privacy Constraints in Federated Learning

**Justin S. Kang**                                          *justin_kang@berkeley.edu*
*UC Berkeley*

**Ramtin Pedarsani**                                          *ramtin@ece.ucsb.edu*
*UC Santa Barbara*

**Kannan Ramchandran**                                          *kannanr@berkeley.edu*
*UC Berkeley*

**Reviewed on OpenReview:** *https://openreview.net/forum?id=ynG5Ak7n7Q*

## Abstract

Modern data aggregation often involves a platform collecting data from a network of users with various privacy options. Platforms must solve the problem of how to allocate incentives to users to convince them to share their data. This paper puts forth an idea for a *fair* amount to compensate users for their data at a given privacy level based on an axiomatic definition of fairness, along the lines of the celebrated Shapley value. To the best of our knowledge, these are the first fairness concepts for data that explicitly consider privacy constraints. We also formulate a heterogeneous federated learning problem for the platform with privacy level options for users. By studying this problem, we investigate the amount of compensation users receive under fair allocations with different privacy levels, amounts of data, and degrees of heterogeneity. We also discuss what happens when the platform is forced to design fair incentives. Under certain conditions we find that when privacy sensitivity is low, the platform will set incentives to ensure that it collects all the data with the lowest privacy options. When the privacy sensitivity is above a given threshold, the platform will provide no incentives to users. Between these two extremes, the platform will set the incentives so some fraction of the users chooses the higher privacy option and the others chooses the lower privacy option.

## 1 Introduction

From media to healthcare to transportation, the vast amount of data generated by people every day has solved difficult problems across many domains. Nearly all machine learning algorithms, including those based on deep learning rely heavily on data and many of the largest companies to ever exist center their business around this precious resource of data. This includes directly selling access to data to others for profit, selling targeted advertisements based on data, or by exploiting data through data-driven engineering, to better develop and market products. Simultaneously, as users become more privacy conscious, online platforms are increasingly providing *privacy level* options for users. Platforms may provide incentives to users to influence their privacy decisions. This manuscript investigates how platforms can fairly compensate users for their data contribution at a given privacy level. Consider a platform offering geo-location services with three privacy level options:

   i) Users send no data to the platform — all data processing is local and private.

   ii) An intermediate option with federated learning (FL) for privacy. Data remains with the users, but the platform can ask for gradients with respect to a particular loss function.

   iii) A non-private option, where all user data is stored and owned by the platform.

If users choose option (i), the platform does not stand to gain from using that data in other tasks. If the user chooses (ii), the platform is better off, but still has limited access to the data via FL and may not be able to fully leverage its potential. Therefore, the platform wants to incentivize users to choose option (iii). This may be done by providing services, discounts or money to users that choose this option. Effectively, by choosing an option, users are informally selling (or not selling) their data to platforms. Due to the lack of a formal exchange, it can be difficult to understand if this sale of user data is *fair*. Are platforms making the cost of choosing private options like (i) or (ii) too high? Is the value of data much higher than the platform is paying?

A major shortcoming of the current understanding of data value is that it often fails to explicitly consider a critical factor in an individual's decision to share data—privacy. This work puts forth two rigorous notions of the fair value of data in Section 3 that explicitly include privacy and make use of the axiomatic framework of the *Shapley value* from game theory (Shapley, 1952).

Compelled by the importance of data in our modern economy and a growing social concern about privacy, this paper presents frameworks for quantifying the fair value of private data. Specifically, we consider a setting where users are willing to provide their data to a platform in exchange for some sort of payment and under some privacy guarantees depending on their level of privacy requirements. The platform is responsible for running the private learning algorithm on the gathered data and making the fair payments with the objective of maximizing its utility including statistical accuracy and total amount of payments. Our goal is to understand fair mechanisms for this procedure as depicted in Fig. 1.

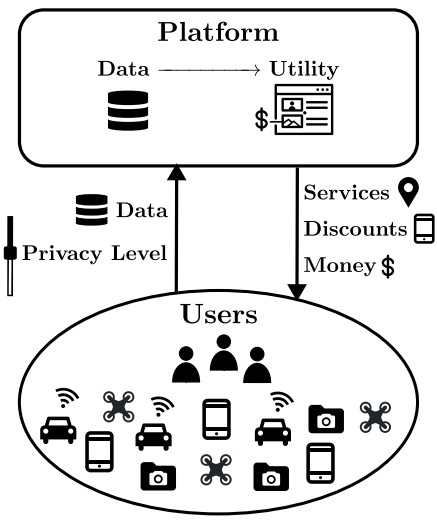

Figure 1: Depiction of interactions between platform and users. Users generate data with phones, cameras, vehicles, and drones. This data goes to the platform but requires some level of privacy. The platform uses this data to generate utility, often by using the data for learning tasks. In return, the platform may provide the users with payments in the form of access to services, discounts on products, or monetary compensation.

## 1.1 Related Work

**Economics**   With the widespread use of the internet, interactions involving those that have data and those that want it have become an important area of study (Balazinska et al., 2011), and a practical necessity (Spiekermann et al., 2015b). Among these interactions, the economics of data from privacy conscious users has received significant attention in Acquisti et al. (2016) and Wieringa et al. (2021). The economic and social implications of privacy and data markets are considered in Spiekermann et al. (2015a). In Acemoglu et al. (2019) the impact of data externalities is investigated. The leakage of data leading to the suppression of its market value is considered.

**Privacy**   Currently, popular forms of privacy include federated learning (Kairouz et al., 2021) and differential privacy (DP) (Dwork, 2008; Bun & Steinke, 2016) either independently or in conjunction with one another. Our work uses a flexible framework that allows for a rage of different privacy models to be considered.

**Optimal Data Acquisition**   One line of literature studies *data acquisition*, where platforms attempt to collect data from privacy conscious users. Ghosh & Ligett (2013) consider the case of uniform privacy guarantees (homogeneous DP), where users have unique minimum privacy constraints, focusing on characterizing equilibria. Ghosh & Roth (2011) allows for heterogeneous DP guarantees with the goal to design a dominant strategy truthful mechanism to acquire data and estimate the sum of users' binary data. In Fallah et al. (2022) the authors consider an optimal data acquisition problem in the context of private mean estimation in two different local and central heterogeneous DP settings. It is assumed that players care about both the estimation error of the common estimator generated by the platform and any payments made to them by the platform in their decision making. By assuming linear privacy sensitivity represented by scalars and drawn from a distribution,

they devise a mechanism for computing the near-Bayes optimal privacy levels to provide to the players. Cummings et al. (2023) focuses on the central setting, under both the linear privacy sensitivity and the privacy constraints model, offering insights into the optimal solution. Hu & Gong (2020) goes beyond linear estimation to consider FL, where each user has a unique privacy sensitivity function parameterized by a scalar variable. Users choose their privacy level, and the platform pays them via a proportional scheme. For linear privacy sensitivity functions, an efficient way to compute the Nash equilibrium is derived. Roth & Schoenebeck (2012); Chen et al. (2018); Chen & Zheng (2019) also follow Ghosh & Roth (2011) and design randomized mechanisms that use user data with a probability that depends on their reported privacy sensitivity value.

**Fairness** In Jia et al. (2019), Ghorbani & Zou (2019) and Ghorbani et al. (2020) a framework for determining the fair value of data is proposed. These works extend the foundational principles of the Shapley value (Shapley, 1952), which was originally proposed as a concept for utility division in coalitional games to the setting of data. Our work takes this idea further and explicitly includes privacy in the definition of the fair value of data, ultimately allowing us to consider private data acquisition in the context of fairness constraints. Finally, we note that we consider the concept of fairness in data valuation, not algorithmic fairness, which relates to the systematic failure of machine learning systems to account for data imbalances.

## 1.2 Main Contributions

- We present an axiomatic notion of fairness that is inclusive of the platforms and the users in Theorem 1. The utility to be awarded to each user and the platform is uniquely determined, providing a useful benchmark for comparison.

- In the realistic scenario that fairness is considered between users, Theorem 2 defines a notion of fairness based on axioms, but only places restriction on relative amounts distributed to the players. This creates an opportunity for the platform to optimize utility under fairness constraints.

- Section 4 contains an example inspired by online platform advertisement to heterogeneous users. We use our framework to fairly allocate payments, noticing how those payments differ among different types of users, and how payments change as the degree of heterogeneity increases or decreases. We numerically investigate the mechanism design problem under this example and see how heterogeneity impacts the optimal behavior of the platform.

- Finally, Section 5 explores the platform mechanism design problem. In Theorem 3 we establish that there are three distinct regimes in which the platform's optimal behavior differs depending on the common privacy sensitivity of the users. While existing literature has investigated how a platform should design incentives for users to optimize its utility, this is the first work to consider fairness constraints on the platform. When privacy sensitivity is low, the platform will set incentives to ensure that it collects all the data with the lowest privacy options. When the privacy sensitivity is above a given threshold, the platform will provide no incentives to users. Between these two extremes, the platform will set the incentives so some fraction of the users chooses the higher privacy option and the remaining chooses the lower privacy option.

**Notation** Lowercase boldface $\mathbf{x}$ and uppercase boldface $\mathbf{X}$ symbols denote vectors and matrices respectively. $\mathbf{X} \odot \mathbf{Y}$ represents the element-wise product of $\mathbf{X}$ and $\mathbf{Y}$. We use $\mathbb{R}_{\geq 0}$ for non-negative reals. Finally, $\mathbf{x} \geq \mathbf{y}$ means that $x_i \geq y_i \ \forall i$. For a reference list of all symbols and their meaning, see Appendix A.

## 2 PROBLEM SETTING

### 2.1 Privacy Levels and Utility Functions

**Definition 1.** A *heterogeneous privacy framework* on the space of random function $A : \mathcal{X}^N \to \mathcal{Y}$ is:

1. A set of *privacy levels* $\mathcal{E} \subseteq \mathbb{R}_{\geq 0} \cup \{\infty\}$, representing the amount of privacy of each user. We use $\rho$ to represent an element of $\mathcal{E}$ in the general case and $\epsilon$ when the privacy levels are referring to DP parameters (defined below).

2. A constraint set $\mathcal{A}(\boldsymbol{\rho}) \subseteq \{A : \mathcal{X}^N \to \mathcal{Y}\}$, representing the set of random functions that respect the privacy levels $\rho_i \in \mathcal{E}$ for all $i \in [N]$. If a function $A \in \mathcal{A}(\boldsymbol{\rho})$ then we call it a $\boldsymbol{\rho}$-private algorithm.

We maintain this general notion of privacy framework because different notions of privacy can be useful in different situations. For example, the lack of rigor associated with notions such as FL, may make it unsuitable for high security applications, but it may be very useful in protecting users against data breaches on servers, by keeping their data local. One popular choice with rigorous guarantees is DP:

**Definition 2.** Pure heterogeneous $\boldsymbol{\epsilon}$-DP, is a heterogeneous privacy framework with $\mathcal{E} = \mathbb{R}_{\geq 0} \cup \{\infty\}$ and the constraint set $\mathcal{A}(\boldsymbol{\epsilon}) = \{A : \Pr(A(\mathbf{x}) \in S) \leq e^{\epsilon_i}\Pr(A(\mathbf{x}') \in S)\}$ for all measurable sets $S$.

Henceforth we will use the symbol $\boldsymbol{\epsilon}$ to represent privacy level when we are specifically referring to DP as our privacy framework, but if we are referring to a general privacy level, we will use $\boldsymbol{\rho}$. Fig. 2, depicts another heterogeneous privacy framework. $\rho_i = 0$ means the user will keep their data fully private, $\rho_i = 1$ is an intermediate privacy option where user data is securely aggregated with other users before it is sent to the platform, which obfuscates it from the platform. Finally, if $\rho_i = 2$, the users send a sufficient statistic for their data to the platform.

The platform applies an $\boldsymbol{\rho}$-private algorithm $A_{\boldsymbol{\rho}} : \mathcal{X}^N \mapsto \mathcal{Y}$ to process the data, providing privacy level $\rho_i$ to data $x_i$. The output of the algorithm $y = A_{\boldsymbol{\rho}}(\mathbf{x})$ is used by the platform to derive utility $U$, which depends on the privacy level $\boldsymbol{\rho}$.

For example, if the platform is estimating the mean of a population, the utility could depend on the mean square error of the private estimator.

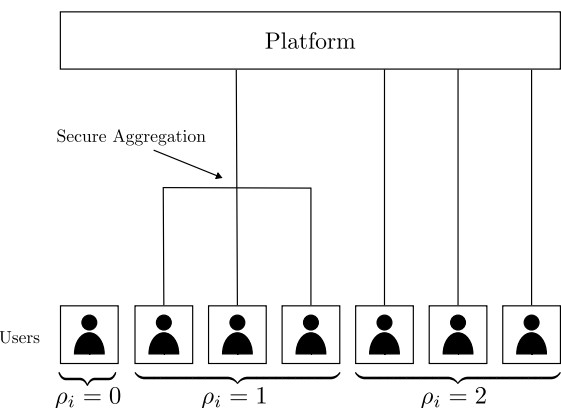

Figure 2: Users choose between three levels of privacy. If $\rho_i = 0$, users send no data to the platform. If $\rho_i = 1$, a user's model is securely combined with other users who also choose $\rho_i = 1$, and the platform receives only the combined model. If $\rho_i = 2$, users send their relevant information directly to the platform.

**Differences from prior work** This formulation differs from the literature of optimal data acquisition (i.e., Fallah et al. (2022)), where *privacy sensitivity* is reported by users, and the platform chooses privacy levels $\rho_i$ based on this sensitivity. Privacy sensitivity is the cost that a user experiences by choosing a particular privacy level. Their formulation allows for a relatively straightforward application of notions like incentive compatibility and individual rationality from mechanism design theory. In this work, we instead emphasize that users choose a privacy level, rather than report a somewhat nebulously defined privacy sensitivity. Despite this difference, the notions of fairness described in the following section can be applied more broadly.

## 2.2 The Data Acquisition Problem

The platform generates transferable and divisible utility $U(\boldsymbol{\rho})$ from the user data. In exchange, distributes a portion of the utility $t_i(\rho_i; \boldsymbol{\rho}_{-i})$ to user $i$, where $\boldsymbol{\rho}_{-i}$ denotes the vector of privacy levels $\boldsymbol{\rho}$ with the $i$th coordinate deleted. These incentives motivate users to lower their privacy level, but each user will also have some sensitivity to their data being shared, modelled by a *sensitivity function* $c_i : \mathcal{E} \to [0, \infty)$, $c_i(0) = 0$. The behavior of users can be modelled with the help of a utility function:

$$u_i(\boldsymbol{\rho}) = t_i(\rho_i, \boldsymbol{\rho}_{-i}) - c_i(\rho_i). \tag{1}$$

The payment to user $i$ will tend to increase with a lower privacy level, as the platform can better exploit the data, but their sensitivity $c_i$ will increase with $\rho_i$, creating a trade-off. By specifying $t_i(\rho_i; \boldsymbol{\rho}_{-i})$, the platform effectively creates a game among the users. This situation is depicted in Fig. 3. Each user's action is the level of privacy that they request for the data they share. Users (players) select their privacy level $\rho_i$

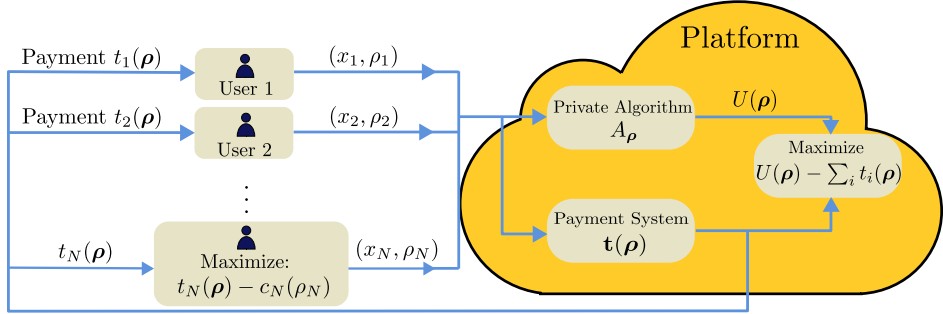

Figure 3: Users send their data $x_i$ and a privacy level $\rho_i$ to the central platform in exchange for payments $t_i(\rho_i; \boldsymbol{\rho}_{-i})$. The central platform extracts utility from the data at a given privacy level and optimizes incentives to maximize the difference between the utility and the sum of payments $U(\boldsymbol{\rho}) - \sum_i t_i(\rho)$.

by considering their utility function $u_i$ and the potential actions of the other players. The platform's goal is to design the payments $t_i(\rho_i; \boldsymbol{\rho}_{-i})$ that maximize its net utility $U(\boldsymbol{\rho}) - \mathbb{1}^T \mathbf{t}(\boldsymbol{\rho})$. One way to formulate this problem is to consider maximizing this difference at equilibrium points:

$$\begin{aligned}
\underset{\mathbf{t}(\cdot), \boldsymbol{\rho}}{\text{maximize}} \quad & U(\boldsymbol{\rho}) - \mathbb{1}^T \mathbf{t}(\boldsymbol{\rho}) \\
\text{subject to} \quad & \boldsymbol{\rho} \in \text{NE}(\mathbf{t}).
\end{aligned} \tag{2}$$

$\text{NE}(\mathbf{t})$ denotes the set of NE strategies induced by the payment function $\mathbf{t}$, which is the vector with payment function $t_i$ at index $i$. Recall that the NE is a stable state such that no user gains by unilaterally changing their strategy. Depending on the circumstances, we may also want to consider equilibrium points in mixed strategies (distributions) over the privacy space. This could make sense if we expect users to continually interact with the platform, making a different choice each time, such that users ultimately converge to their long-run average payoff. In such cases, we can formulate the problem for the platform as:

$$\begin{aligned}
\underset{\mathbf{t}(\cdot), \mathcal{P}}{\text{maximize}} \quad & U(\mathcal{P}) - \mathbb{1}^T \mathbf{t}(\mathcal{P}) \\
\text{subject to} \quad & \mathcal{P} \in \text{NE}(\mathbf{t}).
\end{aligned} \tag{3}$$

where we have used the shorthand $f(\mathcal{P}) = \mathbb{E}_{\boldsymbol{\rho} \sim \mathcal{P}}[f(\boldsymbol{\rho})]$ and $\mathcal{P}$ represents a distribution over the privacy space $\mathcal{E}$. Note that in both equation 3 and equation 2 restrictions must be placed on $\mathbf{t}$, otherwise it can be made arbitrarily negative. *Individual rationality* is a common condition in mechanism design that says that a user can be made no worse off by participation. In Section 5, we consider a fairness constraint.

### 2.3 Model Limitations

**Known sensitivity functions** To solve equation 3, the platform requires the privacy sensitivity $c_i$ of each user, and our solution in Section 5 depends on this information. This can be justified when platforms interact with businesses. For example, an AI heath platform may interact with insurance companies and hospitals and can invest significant resources into studying each of its partners. Another example is advertisement platforms and sellers. Another justification is that the privacy sensitivity $c_i$ is learned by the platforms over time, and we are operating in a regime where the estimates of $c_i$ have converged. An interesting future direction could be investigating this learning problem.

**Data-correlated sensitivity** In Section 5 we treat the sensitivity function $c_i$ as fixed and known, but a practical concern is that $c_i$ may depends on the data $x_i$. Say $x_i$ is biological data pertaining to a disease. Those users with the diseases may have higher $c_i$. Without taking this into account, the collected data will be biased. If our utility function is greatly increased by those users who do have the disease though, they may receive far more payment, compensating for this correlation. We leave a investigation of data-correlated sensitivity and fairness to future work.

**Known transferable and divisible utility** Solving equation 3 also requires knowledge of the utility function. In some cases, the platform may dictate the utility entirely on its own, perhaps to value a diverse set of users. In other cases, like in the estimation setting of Example 3.2, it may represent a more concrete metric, like a risk function that is easily computed. In some cases, however, the utility function may not be easily computed. For example, it may depend on the revenue of a company's product, or the downstream performance of a deep network. We also note that $t_i(\rho_i; \boldsymbol{\rho}_{-i})$ may not represent a monetary transfer. Individuals are often compensated for data via discounts or access to services. A shortcoming of our model is that we assume a divisible and transferable utility, which may fail to capture these nuances of compensation.

**Informed and Strategic Users** We also assume that users can compute and play their equilibrium strategy, which is a standard assumption in game theory. Practically this also means that the platform must be transparent about the incentives, fully publishing this information to the users.

## 3 Axiomatic Fairness with Privacy

What is a fair way to distribute incentives? One approach is to view the users and platforms as a coalition jointly generating utility. Following an axiomatic approach to fairness, the celebrated Shapley value (Shapley, 1952) describes how to fairly divide utility among a coalition. In this section, we take a similar approach to defining fairness. This coalitional perspective is not a complete characterization of the complex dynamics between users and platforms, but we argue that it is still a useful one. One of the benefits of this concept of fairness is that it deals with intrinsic value (i.e., how much of the utility comes from the data). This is in contrast to the *market value* that users are willing to sell for (potentially depressed). This information is particularly useful to economists, regulators, and investors, who are interested in characterizing the value of data as capital for the purposes of analysis, taxation, and investment respectively.

### 3.1 Platform as a Coalition Member

We define a coalition of users and a platform as a collection of $s$ users, with $0 \leq s \leq N$ and up to 1 platform. Let $z \in \{0,1\}$ represent the action of the platform. Let $z = 1$ when the platform chooses to join the coalition, and $z = 0$ otherwise. Let $U(\boldsymbol{\rho})$ be as defined in Section 2. We augment the utility to take into account that the utility is zero if the platform does not participate, and define $\boldsymbol{\rho}_S$ as follows:

$$U(z, \boldsymbol{\rho}) := \begin{cases} U(\boldsymbol{\rho}) & z = 1 \\ 0 & z = 0 \end{cases}, \quad [\boldsymbol{\rho}_S]_i := \begin{cases} \rho_i & i \in S \\ 0 & \text{else} \end{cases}. \tag{4}$$

Let $\phi_p(z, \boldsymbol{\rho})$ and $\phi_i(z, \boldsymbol{\rho})$, $i \in [N]$ represent the "fair" amount of utility awarded to the platform and each user $i$ respectively, given $z$ and $\boldsymbol{\rho}$, otherwise described as the "value" of a user. Note that these values depend implicitly on both the private algorithm $A_{\boldsymbol{\rho}}$ and the utility function $U$, but for brevity, we avoid writing this dependence explicitly. The result of Hart & Mas-Colell (1989) show that these values are unique and well defined if they satisfy the following three axioms:

A.i) **Two equally contributing users should be paid equally**. For any $i, j \in [N] : U(z, \boldsymbol{\rho}_{S \cup \{i\}}) = U(z, \boldsymbol{\rho}_{S \cup \{j\}}) \ \forall S \subset [N] \backslash \{i,j\} \implies \phi_i(z, \boldsymbol{\rho}) = \phi_j(z, \boldsymbol{\rho})$.

In addition, for any user $i \in [N]$, $U(1, \boldsymbol{\rho}_{S \cup \{i\}}) - U(1, \boldsymbol{\rho}_S) = 0 \ \forall S \subset [N] \backslash \{i\} \implies \phi_i(z, \boldsymbol{\rho}) = 0$.

A.ii) **The sum of all payments is the total utility**. The sum of values is the total utility $U(z, \boldsymbol{\rho}) = \phi_p(z, \boldsymbol{\rho}) + \sum_i \phi_i(z, \boldsymbol{\rho})$.

A.iii) **If two utility functions are combined, the payment for the combined task should be the sum of the individual tasks**. Let $\phi_p(z, \boldsymbol{\rho})$ and $\phi_i(z, \boldsymbol{\rho})$ be the value of the platform and users respectively for the utility function $U$, under the $\boldsymbol{\rho}$-private $A_{\boldsymbol{\rho}}$. Let $V$ be a separate utility function, also based on the output of $A_{\boldsymbol{\rho}}$, and let $\phi_p'(z, \boldsymbol{\rho})$ and $\phi_i'(z, \boldsymbol{\rho})$ be the utility of the platform and individuals with respect to $V$. Then under the utility $U + V$, the value of user $i$ is $\phi_i(z, \boldsymbol{\rho}) + \phi_i'(z, \boldsymbol{\rho})$ and the value of the platform is $\phi_p(z, \boldsymbol{\rho}) + \phi_p'(z, \boldsymbol{\rho})$.

**Theorem 1.** *Let $\phi_p(z, \epsilon)$ and $\phi_i(z, \epsilon)$ satisfying axioms* (A.i-iii) *represent the portion of total utility awarded to the platform and each user i from utility $U(z, \epsilon)$. Then they are unique and take the form:*

$$\phi_p(z, \boldsymbol{\rho}) = \frac{1}{N+1} \sum_{S \subseteq [N]} \frac{1}{\binom{N}{|S|}} U(z, \boldsymbol{\rho}_S), \tag{5}$$

$$\phi_i(z, \boldsymbol{\rho}) = \frac{1}{N+1} \sum_{S \subseteq [N] \setminus \{i\}} \frac{1}{\binom{N}{|S|+1}} \left( U(z, \boldsymbol{\rho}_{S \cup \{i\}}) - U(z, \boldsymbol{\rho}_S) \right). \tag{6}$$

Theorem 1 is proved in Appendix B.2, and resembles the classic Shapley value result (Shapley, 1952).

### 3.2 Fairness Among Users

Though we can view the interactions between the platform and the users as a coalition, due to the asymmetry that exists between the platform and the users, it also makes sense to discuss fairness among the users alone. In this case, we can consider an analogous set of axioms that involve only the users.

B.i) **Two equally contributing users should be paid equally**. For any $i, j \in [N] : U(\boldsymbol{\rho}_{S \cup \{i\}}) = U(\boldsymbol{\rho}_{S \cup \{j\}}) \ \forall S \subset [N] \setminus \{i, j\} \implies \phi_i(\boldsymbol{\rho}) = \phi_j(\boldsymbol{\rho})$.

In addition, for any user $i \in [N]$, $U(\boldsymbol{\rho}_{S \cup \{i\}}) - U(\boldsymbol{\rho}_S) = 0 \ \forall S \subset [N] \setminus \{i\} \implies \phi_i(\boldsymbol{\rho}) = 0$.

B.ii) **The sum of all payments is an $\alpha(\epsilon)$ fraction of the total utility**. The sum of values is the total utility $\alpha(\boldsymbol{\rho}) U(\boldsymbol{\rho}) = \sum_i \phi_i(\boldsymbol{\rho})$. Where if $U(\boldsymbol{\rho}) = U(\tilde{\boldsymbol{\rho}})$ then $\alpha(\boldsymbol{\rho}) = \alpha(\tilde{\boldsymbol{\rho}})$ and $0 \leq \alpha(\boldsymbol{\rho}) \leq 1$.

B.iii) **If two utility functions are combined, the payment for the combined task should be the sum of the individual tasks**. Let $\phi_i(\boldsymbol{\rho})$ be the value of users for the utility function $U$, under the $\epsilon$-private algorithm $A_{\boldsymbol{\rho}}$. Let $V$ be a separate utility function, also based on the output of the algorithm $A_\epsilon$, and let $\phi_i'(\boldsymbol{\rho})$ be the utility of the users with respect to $V$. Then under the utility $U + V$, the value of user $i$ is $\phi_i(\boldsymbol{\rho}) + \phi_i'(\boldsymbol{\rho})$.

A notable difference between these axioms and (A.i-iii) is that the efficiency condition is replaced with pseudo-efficiency. Under this condition, the platform may determine the sum of payments awarded to the players, but this sum should in general depend only on the utility itself, and not on how that utility is achieved.

**Theorem 2.** *Let $\phi_i(\boldsymbol{\rho})$ satisfying axioms* (B.i-iii) *represent the portion of total utility awarded to each user i from utility $U(\boldsymbol{\rho})$. Then for $\alpha(\boldsymbol{\rho})$ that satisfies axiom* (B.ii) *$\phi_i$ takes the form:*

$$\phi_i(\boldsymbol{\rho}) = \frac{\alpha(\boldsymbol{\rho})}{N} \sum_{S \subseteq [N] \setminus \{i\}} \frac{1}{\binom{N-1}{|S|}} \left( U(\boldsymbol{\rho}_{S \cup \{i\}}) - U(\boldsymbol{\rho}_S) \right). \tag{7}$$

The proof of Theorem 2 can be found in Appendix B.2. This result is similar to the classic Shapley value (Shapley, 1952), but differs in its novel asymmetric treatment of the platform.

**Computational Complexity** At first glance it may seem that both notions of fairness have exponential computational complexity of $N |\mathcal{E}|^N$. This is only true for a worst-case exact computation. In order for these notions to be useful in any meaningful way, we must be able to compute them. Thankfully, in practice, $U$ typically has a structure that makes the problem more tractable. In Ghorbani & Zou (2019), Jia et al. (2019), Wang & Jia (2023) and Lundberg & Lee (2017) special structures are used to compute the types of Shapley value sums we are considering with significantly reduced complexity, particularly in cases where the $U$ is related to the accuracy of a deep network. This is critical because we want to compute fair values for large number of users. For example, our platform could be a medical data network with hundreds of hospitals as our users, or a smartphone company with millions of users, and we need to be able to scale computation to accurately compute these fair values.

**Example: Differentially Private Estimation**  In this example, we use DP as our heterogeneous privacy framework. Let $X_i$ represent independent and identically distributed data of user $i$ respectively, with $\Pr(X_i = 1/2) = p$ and $\Pr(X_i = -1/2) = 1 - p$, with $p \sim \text{Unif}(0,1)$. The platform's goal is to construct an $\boldsymbol{\epsilon}$-DP estimator for $\mu := \mathbb{E}[X_i] = p - 1/2$ that minimizes Bayes risk. There is no general procedure for finding the Bayes optimal $\boldsymbol{\epsilon}$-DP estimator, so restrict our attention to $\boldsymbol{\epsilon}$-DP linear-Laplace estimators of the form:

$$A(\mathbf{X}) = \mathbf{w}(\boldsymbol{\epsilon})^T \mathbf{X} + Z, \tag{8}$$

where $Z \sim \text{Laplace}(1/\eta(\boldsymbol{\epsilon}))$. In Fallah et al. (2022) the authors argue that unbiased linear estimators are nearly optimal in a minimax sense for bounded random variables. We assume a squared error loss $L(a, \mu) = (a - \mu)^2$ and let $\mathcal{A}_{\text{lin}}(\boldsymbol{\epsilon})$ be the set of $\boldsymbol{\epsilon}$-DP estimators satisfying equation 8. Then, we define:

$$A_{\boldsymbol{\epsilon}} = \underset{A \in \mathcal{A}_{\text{lin}}(\boldsymbol{\epsilon})}{\arg\min} \; \mathbb{E}[L(A(\mathbf{X}), \mu)] \qquad r(\boldsymbol{\epsilon}) = \mathbb{E}[L(A_{\boldsymbol{\epsilon}}(\mathbf{X}), \mu)]. \tag{9}$$

In words, $A_{\boldsymbol{\epsilon}}$ is an $\boldsymbol{\epsilon}$-DP estimator of the form equation 8, where $\mathbf{w}(\boldsymbol{\epsilon})$ and $\eta(\boldsymbol{\epsilon})$ are chosen to minimize the Bayes risk of the estimator, and $r(\boldsymbol{\epsilon})$ is the risk achieved by $A_{\boldsymbol{\epsilon}}$. Since the platform's goal is to accurately estimate the mean of the data, it is natural for the utility $U(\boldsymbol{\epsilon})$ to depend on $\boldsymbol{\epsilon}$ through the risk function $r(\boldsymbol{\epsilon})$. Note that if $U$ is monotone decreasing in $r(\boldsymbol{\epsilon})$, then $U$ is monotone increasing in $\boldsymbol{\epsilon}$. Let us now consider the case of $N = 2$ users, choosing from an action space of $\mathcal{E} = \{0, \epsilon'\}$, for some $\epsilon' > 0$. Furthermore, take $U$ to be an affine function of $r(\boldsymbol{\epsilon})$: $U(\boldsymbol{\epsilon}) = c_1 r(\boldsymbol{\epsilon}) + c_2$. For concreteness, take $U(\mathbf{0}) = 0$ and $\sup_{\boldsymbol{\epsilon} \in \mathbb{R}} U(\boldsymbol{\epsilon}) = 1$. Note that this ensures that $U$ is monotone increasing in $\boldsymbol{\epsilon}$, and is uniquely defined. Considering the example of a binary privacy space $\mathcal{E} = \{0, \infty\}$ ($\epsilon' = \infty$), the utility can be written in matrix form as:

$$\mathbf{U} = \begin{bmatrix} U([0,0]) & U([0,\epsilon']) \\ U([\epsilon',0]) & U([\epsilon',\epsilon']) \end{bmatrix} = \begin{bmatrix} 0 & 2/3 \\ 2/3 & 1 \end{bmatrix}. \tag{10}$$

Derivations are available in Appendix B.1. Note from equation 5 and equation 6, it is clear that $\phi_p(0, \boldsymbol{\epsilon}) = \phi_i(0, \boldsymbol{\epsilon}) = 0$. Let $\boldsymbol{\Phi}_p$ and $\boldsymbol{\Phi}_i^{(1)}$ represent the functions $\phi_p(1, \boldsymbol{\epsilon})$ and $\phi_i(1, \boldsymbol{\epsilon})$ in matrix form akin to $\mathbf{U}$. Then using equation 5 and equation 6, we find that the fair allocations of the utility are given by:

$$\boldsymbol{\Phi}_p = \begin{bmatrix} 0 & 1/3 \\ 1/3 & 5/9 \end{bmatrix}, \; \boldsymbol{\Phi}_1^{(1)} = \begin{bmatrix} 0 & 1/3 \\ 0 & 2/9 \end{bmatrix}, \; \boldsymbol{\Phi}_2^{(1)} = \begin{bmatrix} 0 & 0 \\ 1/3 & 2/9 \end{bmatrix}. \tag{11}$$

Consider the utility function defined in equation 10, for the $N = 2$ user mean estimation problem with $\mathcal{E} = \{0, \infty\}$. By Theorem 2 the fair allocation satisfying (B.i-iii) must be of the form:

$$\boldsymbol{\Phi}_1^{(2)} = \mathbf{A} \odot \begin{bmatrix} 0 & 2/3 \\ 0 & 1/2 \end{bmatrix}, \quad \boldsymbol{\Phi}_2^{(2)} = \mathbf{A} \odot \begin{bmatrix} 0 & 0 \\ 2/3 & 1/2 \end{bmatrix}, \; \mathbf{A} = \mathbf{A}^T, \; 0 \leq [\mathbf{A}]_{ij} \leq 1. \tag{12}$$

## 4   Fair Incentives in Federated Learning

FL is a distributed learning process used when data is either too large or too sensitive to be directly transferred in full to the platform. Instead of combining all the data together and learning at the platform, each user performs some part of the learning locally and the results are aggregated at the platform, providing some level of privacy. Donahue & Kleinberg (2021) consider a setting where heterogeneous users voluntarily opt-in to federation. A natural question to ask is: how much less valuable to the platform is a user that chooses to federate with others as compared to one that provides full access to their data? Furthermore, how should the platform allocate incentives to get users to federate? This section addresses these questions.

Each user $i \in [N]$ has a unique mean and variance $(\theta_i, \sigma_i^2) \sim \Theta$, where $\Theta$ is some global joint distribution. Let $\theta_i$ represent some information about the user critical for advertising. We wish to learn $\theta_i$ as accurately as possible to maximize our profits, by serving the best advertisements possible to each user. User $i$ draws $n_i$ samples i.i.d. from its local distribution $\mathcal{D}_i(\theta_i, \sigma_i^2)$, that is, some distribution with mean $\theta_i$ and variance

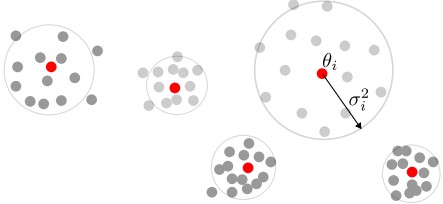

Figure 4: Each user $i \in [N]$ has mean and variance $(\theta_i, \sigma_i^2) \sim \Theta$, where $\Theta$ is a global joint distribution. Let $s^2 = \text{Var}(\theta_i)$ and $r^2 = \mathbb{E}[\sigma_i^2]$ for all $i$. In this case $s^2$ is large relative to $r^2$, and the data is very heterogeneous.

$\sigma_i^2$. Let $s^2 = \text{Var}(\theta_i)$ represent the variance between users and $r^2 = \mathbb{E}[\sigma_i^2]$ represent the variance within a user's data. When $s^2 \gg r^2/n_i$ the data is very heterogeneous, and it is generally not helpful to include much information from the other users when estimating $\theta_i$, however, if $s^2 \ll r^2/n_i$, the situation is reversed, and information from the other users will be very useful. The goal of the platform is to construct estimators $\hat{\theta}_i^p$ while respecting the privacy level vector $\boldsymbol{\rho}$:

$$\text{EMSE}_i(\boldsymbol{\rho}) := \mathbb{E}\left[\left(\hat{\theta}_i^p(\boldsymbol{\rho}) - \theta_i\right)^2\right]. \tag{13}$$

Fig. 2 summarizes our FL formulation. Users can choose from a 3-level privacy space $\mathcal{E} = \{0, 1, 2\}$. In this case the privacy space is not related to DP, but instead encodes how users choose to share their data with the platform. Let $N_j$ be the number of users that choose privacy level $j$. The *heterogeneous privacy framework* is given in Table 1. Note that the error in estimating $\theta_i$ depends not just on the privacy level of the $i$th

| Level | Description | Platform gets |
|---|---|---|
| $\rho_i = 2$ | Provide local estimator directly to the platform. | $\hat{\theta}_i$ |
| $\rho_i = 1$ | Provide securely aggregated model with other users of same privacy level. | $\hat{\theta}^f = \frac{1}{N_1}\sum_{i:\rho_i=1}\hat{\theta}_i$ |
| $\rho_i = 0$ | Provide no data to the platform. | Nothing |

Table 1: Privacy Level Description

user $\rho_i$, but on the entire privacy vector. Let the users be ordered such that $\rho_i$ is a non-increasing sequence. Then for each $i$ the platform constructs estimators of the form:

$$\hat{\theta}_i^p = w_{i0}\hat{\theta}^f + \sum_{j=1}^{N_2} w_{ij}\hat{\theta}_j, \tag{14}$$

where, $\sum_j w_{ij} = 1$ for all $i$. In Proposition 5, found in Appendix B.3, we calculate the optimal choice of $w_{ij}$ which depends on $\boldsymbol{\rho}$. From these estimators, the platform generates utility $U(\boldsymbol{\rho})$. The optimal $w_{i0}$ and $w_{ij}$ in equation 14 are well defined in a Bayesian sense if $\rho_i > 0$ for some $i$, but this does not make sense when $\boldsymbol{\rho} = \mathbf{0}$. We can get around this by defining $\text{EMSE}_i(\mathbf{0}) := r^2 + 2s^2$. For the purposes of our discussion, we assume a logarithmic form utility function. This logarithmic form is common in utility theories dating back at least to Kelly (1956). In the following section, we make a *diminishing returns* assumption to derive our theoretical result, which the logarithmic utility satisfies. The exact utility function we consider is:

$$U(\boldsymbol{\rho}) := \sum_{i=1}^n a_i \log\left(\frac{(r^2 + 2s^2)}{\text{EMSE}_i(\boldsymbol{\rho})}\right). \tag{15}$$

$a_i$ **represents the relative importance of each user**. This is important to model because some users may spend more than others, and are thus more important to the platform i.e., the platform may care about computing their $\theta_i$ more accurately than the other users. The argument of the log is increasing as the EMSE decreases. The log means there is diminishing returns as each $\hat{\theta}_i$ becomes more accurate.

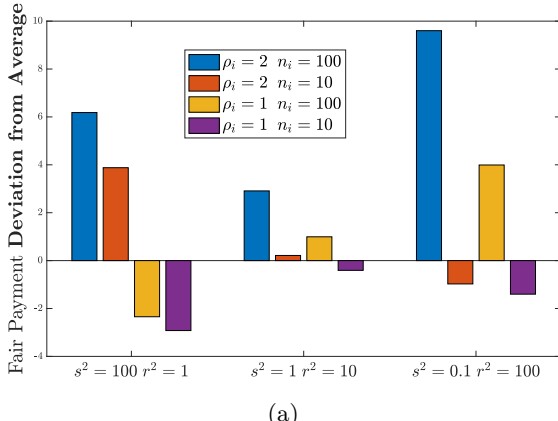
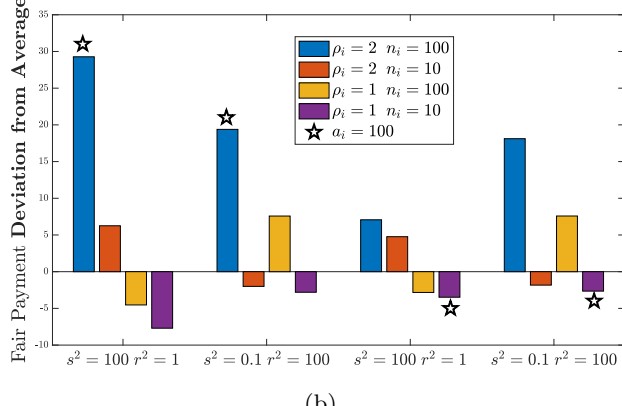

Figure 5: **(a)** Plot of *difference from the average* utility per user $U(\boldsymbol{\rho})/N$ for each of the four different types of users, for three different regimes of $s^2 = \text{Var}(\theta_i)$ and $r^2 = \mathbb{E}[\sigma_i^2]$, with heterogeneity decreasing from left to right. In left (most heterogeneous) plot users who choose $\rho_i = 2$ are more valuable compared to those that choose $\rho_1 = 1$. In the center there is an intermediate regime, where all users are paid closer to the average, with users with more data being favored slightly. In the rightmost graph, with little heterogeneity users with more data are paid more, and privacy level has a lesser impact on the payments.
**(b)** In each case there is one user $i$ with $a_i = 100$ (indicated with a star), while all other users $j \neq i$ have $a_j = 1$ ( $a_i$ represents the relative importance of the user in the utility function). In the two leftmost set of bars, we see that the user with $\rho_i = 2$ and $n_i = 100$ receives by far the most payment, when heterogeneity is high, but this becomes less dramatic as heterogeneity decreases. This shows that when users are very heterogeneous, if $a_i$ is large for only user $i$, most of the benefit in terms of additional payments should go to user $i$. Likewise, comparing the second from the left and the rightmost plots we see little difference, showing that the opposite is true in the homogeneous case: any user can benefit from any other user having a large $a_i$.

## 4.1 Fair Payments Under Optional Federation

In this section, we focus on our definition of fairness in Theorem 2 and analyze the fair values $\phi_i(\boldsymbol{\rho})$ that are induced when using that notion of fairness. Let there be $N = 10$ users. $N_1 = 5$ of these users opt for federating ($\rho_i = 1$), $N_2 = 4$ directly provide their data to the platform ($\rho_i = 2$), and finally, $N_0 = 1$ user chooses to not participate ($\rho_i = 0$). In this subsection (and Fig. 5), without loss of generality, we assume $\alpha(\boldsymbol{\rho}) = 1$, and the results of this section can be scaled accordingly. The choices of $\rho_i$ for each user depends on their individual privacy sensitivity functions $c_i$, but we defer that discussion to the next subsection.

**Different Amounts of Data**  Fig 5a plots the difference from an equal distribution of utility, i.e., how much each user's utility differs from $U(\boldsymbol{\rho})/N$. We assume $a_i = 1$ for all users. In the bars furthest to the left, where $s^2 = 100$ and $r^2 = 1$, we are in a very heterogeneous environment. Intuitively, this means that a user $j$ will have data that may not be helpful for estimating $\theta_i$ for $j \neq i$, thus those users that choose $\rho_i = 2$ are paid the most, since at the very least, the information they provide can be used to target their own $\theta_i$. Likewise, users that federate obfuscate where their data is coming from, making their data less valuable (since their own $\theta_i$ cannot be targeted), so users with $\rho_i = 1$ are paid less than an even allocation. On the right side, we have a regime where $s^2 = 0.1$ and $r^2 = 100$, meaning users are similar and user data more exchangeable. Now users with larger $n_i$ are paid above the average utility per user, while those with lower $n_i$ are paid less. Users with $\rho_i = 2$ still receive more than those with $\rho_i = 1$ when $n_i$ is fixed, and this difference is significant when $n_i = 100$. In the center we have an intermediate regime of heterogeneity, where $s^2 = 1$ and $r^2 = 10$. Differences in payments appear less pronounced, interpolating between the two extremes.

**More Valuable Users**  Fig 5b is like Fig 5a, except now in each set of graphs, exactly one user has $a_i = 100$, meaning that estimating $\theta_i$ for user $i$ is 100 times more important than the others. Looking at the two leftmost sets of bars in Fig 5b we see that when user $i$ with $\rho_i = 2$ and $n_i = 100$ is the most important

one, when $s^2$ is large compared to $r^2$, it is user $i$ who receives most of the benefit in terms of its payment but when $s^2$ is smaller, other users also benefit. This can be intuitively explained as follows: if users are very heterogeneous, other users $j \neq i$ do not have data that is helpful for determining $\theta_i$, thus they do not benefit when user $i$ has a larger $a_i$. Likewise, when $s^2$ is small compared to $r^2$ not just user $i$ benefits, but also all those users that contribute more data, as those users with $\rho_i = 1$ and $n_i = 100$ are also paid over the average utility per user. Another key point is the similarity between the second and fourth set of graphs. This tells an interesting story: when users are not very heterogeneous, regardless of which user is has $a_i = 100$, it is those users with large $n_i$ that will benefit.

## 4.2 Platform and User Game - Mechanism Design

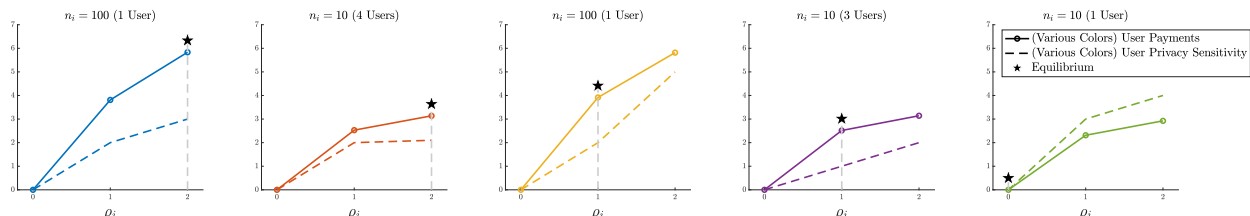

Figure 6: Plot of payments and user sensitivities $c_i$ for 5 different types of users with different sensitivity functions. Some user types are repeated, (indicated in the title), such that there is a total of $N = 10$ Users. Users choose the privacy level $\rho_i$ to maximize the difference between their payment and privacy sensitivity function. A star in each plot marks each user's top choice. The payment functions $\phi_i(\boldsymbol{\rho})$ changes based on the privacy levels of all the users. At the given configuration, which matches the left-most plot in Fig 5b ($s^2 = 100, r^2 = 1$), no user benefits by changing their choice of $\rho_i$, thus the configuration is a Nash Equilibrium.

**Nash Equilibrium Under Fair Payments** We have just discussed how the fair values $\phi_i(\boldsymbol{\rho})$ change depending on the model parameters, and the choices of privacy level $\rho_i$. Now we discuss *how* the users come to decide their $\rho_i$. We again focus on the notion of privacy in Theorem 2, but we now restrict $\alpha(\boldsymbol{\rho}) = \alpha \in [0, 1]$. To avoid overly complicating the model, we take $a_i = 1$ for all users. User $i$ chooses their privacy level $\rho_i$ based on both the payments that they receive from the platform $\alpha\phi_i(\rho_i; \boldsymbol{\rho}_{-i})$ as well as their own unique privacy sensitivity function $c_i(\rho_i)$. In this context, we can view each user as optimizing their own local utility function:

$$u_i(\rho_i, \boldsymbol{\rho}_{-i}) := \alpha\phi(\rho_i; \boldsymbol{\rho}_{-i}) - c_i(\rho_i). \tag{16}$$

User $i$ individually optimizes this function to determine their *best response* (denoted $\mathrm{BR}_i$) to the choices of the other users and the platform:

$$\mathrm{BR}_i(\boldsymbol{\rho}_{-i}, \alpha) := \arg\max_{\rho_i} u_i(\rho_i, \boldsymbol{\rho}_{-i}) \tag{17}$$

The full *best response function* $\mathrm{BR}(\boldsymbol{\rho}, \alpha) := [\mathrm{BR}_1(\boldsymbol{\rho}_{-1}, \alpha), \ldots, \mathrm{BR}_N(\boldsymbol{\rho}_{-N}, \alpha)]^T$ is a vector that collects all users' individual best response functions. The "fixed points" of the best response at a given $\alpha$ constitute the pure-strategy NEs:

$$\mathrm{NE}(\alpha) := \{\boldsymbol{\rho} : \boldsymbol{\rho} = \mathrm{BR}(\boldsymbol{\rho}, \alpha)\}. \tag{18}$$

Fig 6 depicts a particular fixed point of the best response function (therefore a NE) in our federated mean estimation example. Each of the $N = 10$ users in assigned one of 5 unique sensitivity functions $c_i$, which are shown in the figure.

**Solving the Fair Data Acquisition Problem** The platform's goal is to set $\alpha$ such that it maximizes the total amount of utility it receives. Since the NE set depends on $\alpha$ the platform has some control over the behavior of the users. The mechanism design problem in this case reduces to:

$$
\begin{aligned}
\underset{\alpha, \boldsymbol{\rho}}{\text{maximize}} \quad & (1 - \alpha)U(\boldsymbol{\rho}) \\
\text{subject to} \quad & \boldsymbol{\rho} \in \mathrm{NE}(\alpha).
\end{aligned}
\tag{19}
$$

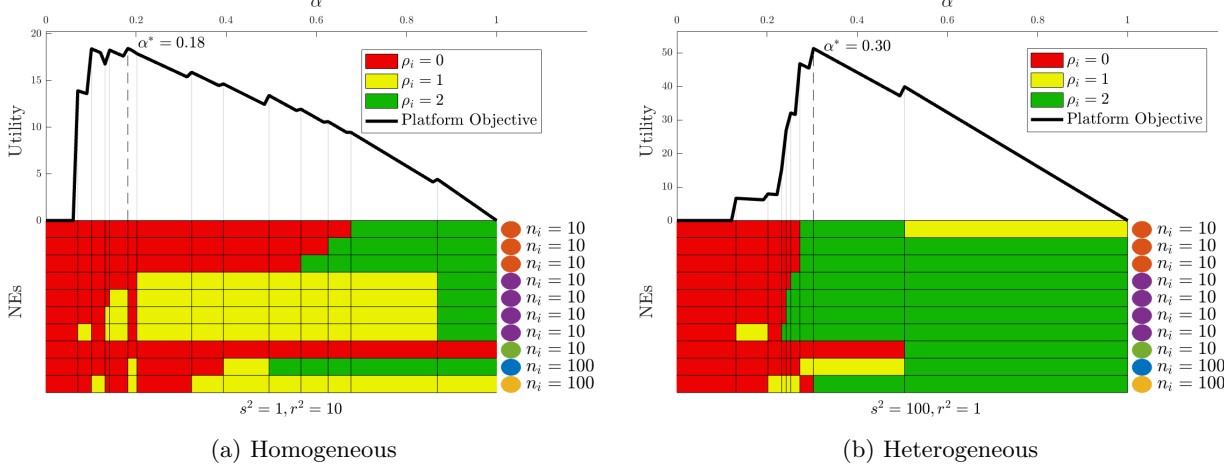

(a) Homogeneous $\qquad\qquad$ (b) Heterogeneous

Figure 7: **(a) and (b)** The top of both sub-figures plot the optimization landscape for the platform design problem equation 19.The solid black line depicts the partial solution to equation 19, where we optimize over $\boldsymbol{\rho}$ for fixed $\alpha$. Precisely that is: $\max_{\boldsymbol{\rho}}(1-\alpha)U(\boldsymbol{\rho})$, $\boldsymbol{\rho} \in \mathrm{NE}(\alpha)$. The bottom of each plot depicts the optimizing $\arg\max_{\boldsymbol{\rho}}(1-\alpha)U(\boldsymbol{\rho})$, $\boldsymbol{\rho} \in \mathrm{NE}(\alpha)$. $\rho_i = 0$ means that the users send no data to the platform, $\rho_i = 1$ means the users securely aggregate their data with other users, and $\rho_i = 2$ means the users send all their data to the platform. Each row of bars corresponds to a unique user $i$. On the right-hand side, the amount of data each user has $n_i$ is given, as well as the color-code corresponding to the $c_i$ of each user. By looking at the same color plot in Fig 6, the $c_i$ for the given user can be found. If there are multiple NEs that achieve the maximum utility, one is shown arbitrarily.
**(a)** Is in a homogeneous regime. In this example it turns out that the optimal $\alpha^*$ for the platform is such that both of the users with $n_i = 100$ choose $\rho_i = 1$, while the other users choose $\rho_i = 0$, showing the importance of the amount of data, as opposed to higher privacy options.
**(b)** Is in a more heterogeneous setting. We find that the optimal $\alpha^*$ ensures that most users choose $\rho_i = 2$, except one user with high privacy sensitivity that chooses $\rho_i = 0$, and one user that chooses $\rho_i = 1$.

Note that this is exactly equation 2, except we have used the fact that $\alpha(\boldsymbol{\rho})U(\boldsymbol{\rho}) = \sum_i \phi_i(\boldsymbol{\rho})$. Solving equation 19 is a daunting task, due to the complex structure of the constraint, however, it is numerically tractable in this case. The key idea to efficiently solve equation 19 is to exploit symmetries in the utility function. For instance, if $\boldsymbol{\rho}'$ is constructed from $\boldsymbol{\rho}$ by permuting the values $\rho_i, \rho_j$ where $n_i = n_j$, then $U(\boldsymbol{\rho}) = U(\boldsymbol{\rho}')$. This allows us compute $\phi_i(\boldsymbol{\rho})$ for a much smaller number of representative $\boldsymbol{\rho}$, rather than a full combinatorial search. Despite producing the same utility, $\boldsymbol{\rho}$ and $\boldsymbol{\rho}'$ may have very different stability properties. To deal this this, we can implement an efficient tree search to identify the NEs within each group. Once we have an efficient algorithm for computing NEs, we conduct a grid search to determine the optimal $\alpha$. Note that for an arbitrary problem, finding the equilibria can be a challenging task, and thus the more difficult it is to compute the equilibria, the more difficult it will be to solve the design problem in equation 2. We provide a full description of our solution in Appendix D.

Fig 7 shows the numerical solution to equation 19 for two different choices of $s^2$ and $r^2$. Fig 7a is a more homogeneous setting and we observe that the optimal $\alpha^*$ sets the payment just high enough so that both of the users with $n_i = 100$ choose to participate at $\rho_i = 1$, while all other users choose $\rho_i = 0$. Essentially, the platform identifies the two users with $n_i = 100$, and focused on collecting their data, rather than the data of the other 8 users. This configuration collects a majority of the data for relatively little payment. Since the data is very homogeneous anyways, the platform is not losing much utility by allowing those two users to choose $\rho_i = 1$ rather than $\rho_i = 2$. As the platform increases $\alpha$, more users choose to participate, eventually with many of them choosing $\rho_i = 2$, however, the benefit is minimal, and it is outweighed by the extra payments that are required to achieve that outcome. Fig 7b covers a more heterogeneous setting. In this setting, we see that $\alpha^*$ is higher, and a larger fraction of the total utility is paid to users. For this $\alpha^*$ only two of the users do not choose $\rho_i = 2$. There is also one user with $\rho_i = 1$, and one with $\rho_i = 0$ that have

a higher privacy sensitivities. In the more heterogeneous setting, allowing users to choose a private option is more costly, so the extra payment to ensure more user choose $\rho_i = 2$ is worthwhile in this regime.

**Other Formulations**   We conclude this section by remarking that this is just one particular formulation of a federated learning problem where users have privacy choice. Another interesting formulation can be found in Aldaghri et al. (2023), which comes from the literature on personalized federated learning Li et al. (2021). In this formulation, all users join the federated learning process, but some users can choose between a private option, where their data is protected via differential privacy, or a standard non-private option. Exploring fair incentives in this, and other models could be an interesting direction for future work.

# 5  Fairness Constraints: Data Acquisition

In the previous section, we considered a complex model, numerically studying the equilibria of the users based on fair payments from the platform, as well as how the platform optimally chooses $\alpha$. In this section, we consider a tractable model and theoretically study how the optimal $\alpha^*$ changes based on the privacy sensitivity of users, under a fairness framework based on Theorem 2. This section addresses this problem by investigating the incentives of a platform designing a mechanism under the constraint of fairness.

Consider $N \geq 2$ users each with identical statistical marginal contribution, i.e., for any $i, j$ we have $S \subseteq [N]\backslash\{i, j\}$, $U(\boldsymbol{\rho}_{S\cup\{i\}}) = U(\boldsymbol{\rho}_{S\cup\{j\}})$. The platform is restricted to making fair payments satisfying axioms (B.i-iii) with the additional constraint that $\alpha(\boldsymbol{\rho}) = \alpha \in [0, 1]$. Users choose one of two available privacy levels $\rho_i \in \mathcal{E}^N$, with $\mathcal{E} = \{\rho'_1, \rho'_2\}$ and $\rho'_2 > \rho'_1$. We can write the utility of the user $i$ as

$$u(\rho_i, \boldsymbol{\rho}_{-i}) = \alpha\phi(\rho_i; \boldsymbol{\rho}_{-i}) - c\mathbb{1}\{\rho_i = \rho'_2\}. \tag{20}$$

Users gain utility from incentives provided by the platform but incur a cost of $c$ if they choose the less private option. For now, we assume $c$ is the same for all users; later we discuss the case where $c$ is different. Note that we can drop the index of $\phi_i$ due to the assumption of equal marginal contribution. To enrich the problem, we allow users to employ a mixed strategy denoted by $\mathbf{p} = [p, (1 - p)]^T$, where users choose the $\rho'_1$ with probability $p$ and $\rho'_2$ with probability $1 - p$. This is justified because we expect users to repeatedly interact with platforms and sample from their mixed strategy and ultimately converge to their expected utility.

The platform is also trying to maximize the fraction of the total expected utility $U(\mathbf{p}) := \mathbb{E}_{\boldsymbol{\rho}\sim\mathbf{p}}[U(\boldsymbol{\rho})]$ that it keeps as in equation 3. The platform's goal is to choose a payment value $\alpha$ such that it optimizes:

$$\begin{aligned}\underset{\alpha}{\text{maximize}} \quad & (1 - \alpha)U(\mathbf{p}^*(\alpha)) \\ \text{subject to} \quad & \mathbf{p}^*(\alpha) \in \text{NE}(\alpha).\end{aligned} \tag{21}$$

The objective is simplified compared to equation 3 by exploiting the pseudo-efficiency axiom, which says that the sum of payments is $\alpha$ times the total utility. The constraint in equation 21 implicitly encodes the user behavior governed by equation 20, and will change with the privacy sensitivity $c$. Theorem 3 characterizes the solution of equation 21 for different values of $c$. To make equation 21 amenable to insightful analysis, we make some mild assumptions.

**Assumption 1.** *The utility $U$ is monotone:* $\boldsymbol{\rho}_S^{(2)} \geq \boldsymbol{\rho}_S^{(1)} \implies U(\boldsymbol{\rho}_S^{(2)}) > U(\boldsymbol{\rho}_S^{(1)}) \ \forall S \subseteq [N]$.

**Assumption 2.** *The utility $U$ has diminishing returns. Let $n_{private}(\boldsymbol{\rho}_S)$ represent the number of elements of $i \in S$ such that $\rho_i = \rho'_1$, i.e., the number of users choosing the higher privacy option. Furthermore, define $\Delta_i U(\boldsymbol{\rho}_S) := U(\boldsymbol{\rho}_S^{(i+)}) - U(\boldsymbol{\rho}_S)$, where $\boldsymbol{\rho}_S^{(i+)}$ is equal to $\boldsymbol{\rho}_S$ except $\rho_i^{(i+)} = \rho'_2$. In other words, $\Delta_i U(\boldsymbol{\rho}_S)$ is the marginal increase in utility when the ith user switches to the lower privacy option. Then $U$ satisfies:*

$$n_{private}(\boldsymbol{\rho}_S^{(1)}) \geq n_{private}(\boldsymbol{\rho}_S^{(2)}) \implies \Delta_i U(\boldsymbol{\rho}^{(1)}) > \Delta_i U(\boldsymbol{\rho}^{(2)}). \tag{22}$$

It is helpful to define the *expected relative payoff*, where the expectation is taken with respect to the actions of the other players. When all other users choose a mixed strategy $\mathbf{p}$, the expected relative payoff is defined as:

$$\gamma(p) := \phi(\rho'_2; \mathbf{p}) - \phi(\rho'_1; \mathbf{p}) = \mathbb{E}_{\substack{\rho_j \sim \mathbf{p} \\ j \neq i}}[\phi(\rho'_2; \boldsymbol{\rho}_{-i}) - \phi(\rho'_1; \boldsymbol{\rho}_{-i})]. \tag{23}$$

For convenience, we have defined $\gamma$ in terms of the scalar $p$, rather than the vector $\mathbf{p} = [p,\ (1-p)]^T$. This quantity represents the expected gain in incentive (normalized to make it invariant to $\alpha$) if a user switches to a less private level from the more private level given everyone else plays the mixed strategy $\mathbf{p}$.

**Theorem 3.** *Consider a binary privacy level game with $N$ users and a platform. If $U$ satisfies Assumptions 1 and 2, and the platform payments are fair as defined in Theorem 2 with constant $\alpha$ then the optimal $\alpha^*$ can be divided into three regimes depending on $c$. The boundaries of these regions are $\gamma_{max} := \max_p \gamma(p)$ and some $c_{th} < \gamma_{max}$ such that:*

1. *When $c > \gamma_{max}$, $\alpha^* = 0$ is the maximizer of 21.*
2. *When $c_{th} < c < \gamma_{max}$ then $\alpha^*$ is the minimizing $\alpha \in [0,1]$ such that $p^*(\alpha) \in \gamma^{-1}(c/\alpha)$.*
3. *When $c < c_{th}$: $\alpha^*$ is the smallest $\alpha \in [0,1]$ such that $p(\alpha) = 0$, where*

$$c_{th} = \max \left\{ c \left| \frac{1 - c/\gamma_{min}}{1 - \alpha} - \frac{U(p^*(\alpha))}{U(0)} \geq 0 \ \forall \alpha \leq c/\gamma_{min} \right. \right\}. \tag{24}$$

Theorem 3 can be interpreted as follows. If privacy sensitivity is above $\gamma_{max}$ for the given task, it is not worth the effort of the platform to participate. On the other hand, if privacy sensitivity is less than $c_{th}$, the platform should set $\alpha$ to be as small as possible, while still ensuring that all users choose the low privacy setting. Finally, if privacy sensitivities lie somewhere in between, $\alpha^*$ should be chosen based on the $\gamma$ function, and generally will lead to a mixed strategy with some proportion of users choosing each of the two options.

**Comparison to other works**   Two key novelties of our work is that we (1) consider a constraint of fairness and (2) have users choose a privacy level, rather than report their privacy sensitivity. This is different from Fallah et al. (2022), and Cummings et al. (2023), which rely on incentive compatibility, and have users report their privacy parameters. In Fallah et al. (2022), a computationally efficient algorithm is proposed for computing user payments and privacy levels to assign users. Both of these works consider a mean estimation problem, where users have i.i.d. samples, and so also have the "equal marginal contribution" assumption that we have. Distinct from our model, users have an additional term in their utility where they benefit from reduced error in the estimation problem. These works focus on maximizing the platform utility, and it is very clear that the payments deviate significantly from the fair ones that satisfy the fairness axioms. Hu & Gong (2020) is perhaps the work most relevant to ours. They consider an incentive design problem where the platform fixes the total sum of payments $R$ and the amount each user receives is proportional to their privacy level $\rho_i$, which the users choose. This proportional scheme, while potentially viewed as a type of fairness, does not satisfy our axioms. For a particular utility function, they develop a computationally efficient algorithm to compute the equilibrium privacy levels $\rho_i$ based on the privacy sensitivities of the users and the total sum of payments $R$. In all of these works, users have a linear privacy sensitivity function with rate $c_i$.

Though this seems different from our binary privacy problem, there is a direct correspondence here since we allow mixed strategies, so in expectation, our sensitivity is also reduced to a linear function of the mixed strategy: i.e., $\mathbb{E}\left[c_i \mathbb{1}\left\{\rho_i = \rho_2'\right\}\right] = c_i \Pr(\rho_i = \rho_2')$.

### 5.1   Mechanism Design: Mean Estimation Example

Let's look at the utility function from equation 11, and fair payments that we calculated from Theorem 2 in equation 10. In this case there are $N = 2$ users, and we will assume each user has a sensitivity function as in equation 20. This setting satisfies the conditions of Theorem 3, and so we can use our above result to characterize the optimal $\alpha^*$ to equation 21 for a range of different $c$ values. That is, as the privacy sensitivity parameter $c$ changes, how is the optimal strategy of the platform impacted? Fig. 8 depicts the solution to equation 21. The

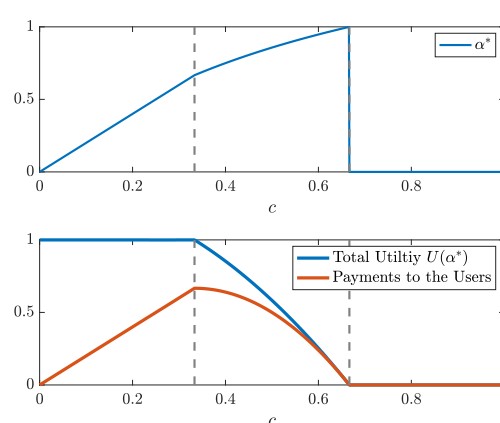

Figure 8: Utility of users and platform when platform solves equation 21. The solution has three separate regions as predicted by Theorem 3.

top plot shows the optimal $\alpha^*$ vs. $c$, while the bottom plot shows the total utility and the fraction of utility paid to each user at the optimal point $\alpha^*$ for a range of $c \in [0, 1]$.

As predicted by Theorem 3, we find that the solution is clearly divided into three regions. Equation 24 tells us that $c_{th} = \frac{1}{3}$ and $\gamma_{max} = \frac{2}{3}$, matching our observations in Fig. 8. In the first region when $c \leq \frac{1}{3}$ the privacy sensitivity of the users is low, and the platform is able to capture most of the utility for itself, paying less of it out to the users. In this region, $\alpha^* = 2c$, growing linearly with the privacy sensitivity. We also see that throughout this regime, the total utility is maximized, as predicted by the theory. In the region where $c \in [\frac{1}{3}, \frac{2}{3}]$, optimal $\alpha^*$ is no longer growing linearly, and now grows as $\alpha^* = \frac{6c}{3c+2}$ no longer have enough incentive to always choose the less private option, the total utility also begins to decrease, meaning less utility is available for incentives. These factors lead to a decrease in total utility in this region. As $\alpha^*$ continues to increase towards 1. Once $\alpha^* = 1$ at $c = \frac{2}{3}$, the platform is getting no utility, so may as well choose $\alpha^* = 0$. Finally, for $c \geq \frac{2}{3}$, the platform no longer attempts to incentivize the users, and the total utility and payments fall to zero with $\alpha^* = 0$.

For this particular example, it is possible to analytically solve the NE constraint in equation 21, exact analytic expressions for the curves in Fig. 8 are given in Appendix B.4.1.

## 5.2 Considering Different Privacy Sensitivities

The computational burden in solving equation 21 is in characterizing the constraint, since the objective reduces to a one-dimensional optimization over $\alpha \in [0, 1]$. In the previous section, with the knowledge that the game is symmetric, we are able to easily characterize the equilibria as a function of $\alpha$. If the $c_i$'s are all different, for arbitrary utility functions, the problem essentially reduces to finding the equilibria in a general game. To make this tractable, we will need some assumptions. In Hu & Gong (2020), the specific choice of utility function and payments makes computation of the equilibrium tractable. If we have only two groups of users with different $c_i$ that act together, and a finite privacy space, we can appeal to tools for enumerating equilibria in matrix games (Avis et al., 2010). In this case if the privacy space is also binary, then the equilibria have an analytical solution, which we provide in Appendix E. Like the symmetric case, there are 3 cases for each of the two users as well as corresponding thresholds that depend on $c_1$ and $c_2$ respectively, resulting in 9 total cases. For example, in the case where payment is below the threshold of both users, neither participate at the low-privacy level, when the payment is high enough both participate at the low privacy level, and for the remaining intermediate cases, either only one user chooses the low privacy option, or there is some asymmetric mixed strategy. Below, we numerically investigate this case:

This problem differs from equation 21 because the equilibrium is governed by asymmetric users. For example, if user 1 and user 2 have privacy sensitivity $c_1$ and $c_2$ respectively, we have

$$u_1(\mathbf{p}_1, \mathbf{p}_2) = \mathbf{p}_1^T \mathbf{\Phi}_1^{(2)} \mathbf{p}_2 - [0 \ c_1]^T \mathbf{p}_1, \quad u_2(\mathbf{p}_1, \mathbf{p}_2) = \mathbf{p}_1^T \mathbf{\Phi}_2^{(2)} \mathbf{p}_2 - [0 \ c_2]^T \mathbf{p}_2. \tag{25}$$

Consider a setting where there are only two users (these can be thought of as representing two *groups* of users) with utility function $u_1$ and $u_2$ listed above. Thus, when the platform is trying to optimize it's own utility, it must take into consideration that these two groups will play different strategies.

$$\begin{aligned} \underset{\alpha}{\text{maximize}} \quad & \mathbf{p}_1^T \mathbf{U} \mathbf{p}_2 - (1 - \alpha)\mathbf{p}_1^T \mathbf{U} \mathbf{p}_2 \\ \text{subject to} \quad & (\mathbf{p}_1, \mathbf{p}_2) \in \text{NE}(\alpha). \end{aligned} \tag{26}$$

Fig. 9 plots the results of simulating the solution of 26. It shows that there is one region when $c_1$ and $c_2$ are both small and close together ($< 1/3$), the platform chooses $\alpha$ to collect data from both users. If the difference is large, even in this region, the users may be asymmetrically engaged. When $c_1 > c_2 > 1/3$, the platform chooses $\alpha$ such that only user 2 chooses to participate, even if the difference is very small, and vice versa if $c_2 > c_1 > 1/3$, as before, when $c_1, c_2 > 2/3$ the sensitivity to too high and the platform can no longer offer enough payment to the users.

**Broader Impact Statement** One of the unique defining characteristics of data is that its generation process is inherently distributed, so no single entity exists to advocate for data sellers. In the past, platforms

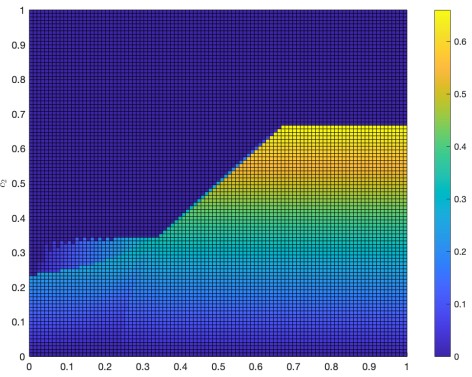 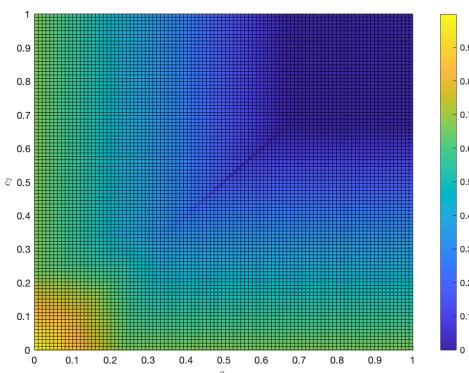

Figure 9: (Left) The payments to user 2 from the platform for a range of $c_1, c_2$. (Right) The platform's share of utility for the optimal $\alpha^*$ payments for a range of values $c_1$, $c_2$.

have been able to extract data from users, often with little to no compensation in return. As public consciousness around privacy changes, a nuanced relationship around privacy between platforms and users must develop. Transparency and understanding the value of user data is an important step in empowering regulators, consumers, and platforms.

- Users making strategic decisions about when they share their data stand to gain from incentives.
- For regulators, understanding the amount of value that flows through the interactions between platforms can enable better policies around data. Frameworks like those discussed in Theorem 1 and 2 can be a starting point in understanding exactly how much this value is.
- For platforms, understanding which data tasks are economically viable, and how they allocate incentive is important. Our discussion in Section 5, and our three regimes help shed light on this.

## 6 Conclusion

This paper introduces two formal definitions of fair payments in the context of acquisition of private data. The first treats the users and the platform together and uses axioms like those of the Shapley value to determine a unique fair distribution of utility. In the second, we define a notion of fairness between the users only, leading to a definition of fairness that admits a range of values, of which the platform is free to choose the most favorable. By formulating a federated mean estimation problem, we show that heterogeneous users can have significantly different contributions to the overall utility, and that a fair incentive, according to our second notion, must take into account the amount of data, privacy level as well as the degree of heterogeneity. We formulate and solve the fairness-constrained mechanism design problem in this federated mean estimation problem, and also find that data heterogeneity and user properties play an important role in the solution.

While previous literature has investigated how platforms should design incentives for users in order to optimize its utility, the definitions of fairness we propose offers another important way to evaluate the fairness of these mechanisms. This is a critical step towards future research in ensuring that data acquisition mechanisms are *both* fair for users and efficient for platforms.

Though we provide a characterization of optimal fair mechanisms when privacy sensitivity is the same across users, designing mechanisms and developing theories that scale up these solutions to deal with platform that interact with large and diverse groups of users is critical. Additionally, users may come and go, as their sensitivities may change over time. Understanding how fluctuating users alter the model is of great practical significance. Furthermore, there is subjectivity in the choice of axioms, and other choices may lead to meaningful notions of fairness worthy of study. We have also assumed a non-divisible and transferable utility, but in many cases, users are paid for their data in the form of access to services. Investigating the impact of this will also be important for the practical application of a comprehensive theory for fairness.

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

## A  Notation Table

| Symbol | Definition | Relevance |
|--------|------------|-----------|
| $\mathcal{E}$ | privacy level space | |
| $N$ | Number of users | |
| $\rho_i$ | privacy level (generic) of user $i$ | |
| $\epsilon_i$ | privacy level (DP only) of user $i$ | |
| $U(\cdot)$ | platform utility function | |
| $u_i(\cdot)$ | user $i$'s utility function | All |
| $c_i(\cdot)$ | privacy sensitivity function | |
| $\mathrm{NE}(\cdot)$ | Nash Equilibrium | |
| $t_i(\cdot)$ | generic payment function | |
| $\phi_i(\cdot)$ | fair payment function for user $i$ under Theorem 1 & 2 | |
| $\phi_p(\cdot)$ | fair payment for platform under Theorem 2 | |
| $\alpha(\cdot)$ | fraction of utility used as payment in Theorem 2 | |
| $n_i$ | user $i$'s amount of data | |
| $s^2$ | variance of user means | |
| $r^2$ | mean of user variances | Section 4 |
| $a_i$ | relative user value in utility function | |
| $\mathrm{BR}_i(\cdot)$ | best response function | |
| $p$ | prob. of playing more private option in mixed strategy | Section 5 |
| $\gamma(\cdot)$ | expected relative payoff by changing strategy | |

Table 2: Table of Common Notations

## B  Missing Proofs

### B.1  Proof of Equation 37

In this section, we present the calculations required to arrive at the utility values in equation 37. First let's treat the trivial case of $\epsilon_1 = 0$, $\epsilon_2 = 0$. The optimal $\epsilon$-DP estimator is simply the optimal Bayes estimator with no data, i.e., the prior mean. Let us define this estimator as $\hat{\mu}_{(0,0)} = 0$. Its risk function is

$$R(\mu, \hat{\mu}_{(0,0)}) = \mathbb{E}\left[L(\hat{\mu}_{(0,0)}, \mu) \mid \mu\right] = \mu^2. \tag{27}$$

The Bayes risk of $\hat{\mu}_{(0,0)}$ is the expectation of this quantity taken using our prior:

$$r([0,0]) = \mathbb{E}\left[\mu^2\right] = \frac{1}{12}. \tag{28}$$

Next, consider the case where user $i$ chooses privacy level $\epsilon_1 = \epsilon' > 0$, and the other user chooses $\epsilon_2 = 0$. In this case the estimator depends on $X_1$, $\hat{\mu}_{(\epsilon',0)} = w_1 X_1 + Z$. Then the risk function is:

$$R(\mu, \hat{\mu}_{(\epsilon',0)}) = \mathbb{E}\left[(w_1 X_1 + Z - \mu)^2 \mid \mu\right] = \left(\mu + \frac{1}{2}\right)\left(\mu - \frac{w_1}{2}\right)^2 + \left(-\mu + \frac{1}{2}\right)\left(\mu + \frac{w_1}{2}\right)^2 + \frac{2}{\eta^2}. \tag{29}$$

Now taking the expectation with respect to our prior over $\mu$, we have:

$$\mathbb{E}\left[R(\mu, \hat{\mu}_{(\epsilon',0)})\right] = \frac{1}{12}\left(3w_1^2 - 2w_1 + 1\right) + \frac{2}{\eta^2}, \tag{30}$$

here $\eta$ is the inverse scale parameter for $Z$. Note that equation 30 is minimized when $\eta$ is maximized. The $\boldsymbol{\epsilon}$-DP condition enforces the constraint $\eta \leq \frac{\epsilon'}{w_1}$. This constraint will be met with equality for the optimal $w_1$. The optimal $w_1^* = \frac{1}{3+\frac{24}{\epsilon'^2}}$. Thus, we have:

$$\hat{\mu}_{(\epsilon',0)} = \frac{1}{3+\frac{24}{\epsilon'^2}}X_1 + Z, \quad Z \sim \text{Laplace}\left(\frac{\epsilon'}{3\epsilon'^2 + 24}\right), \tag{31}$$

and the resulting Bayes risk is:

$$r([\epsilon',0]) = r([0,\epsilon']) = \frac{1}{12}\left(1 - \frac{1}{3+\frac{24}{\epsilon'^2}}\right). \tag{32}$$

For the case with $\epsilon_1 = \epsilon_2 = \epsilon'$ we can repeat the same process by defining $\hat{\mu}_{(\epsilon',\epsilon')} = w_1 X_1 + w_2 X_2 + Z$. By symmetry, we must have $w_1 = w_2$, so we drop the index. Then the risk function and its expectation are:

$$R(\mu, \hat{\mu}_{(\epsilon',\epsilon')}) = 2\left(\mu+\frac{1}{2}\right)\left(-\mu+\frac{1}{2}\right)\mu^2 + \left(\mu+\frac{1}{2}\right)^2(w-\mu)^2 + \left(-\mu+\frac{1}{2}\right)^2(\mu+w)^2 + \frac{2^2}{\eta} \tag{33}$$

$$\mathbb{E}\left[R(\mu, \hat{\mu}_{(\epsilon',\epsilon')})\right] = \frac{1}{12}(8w^2 - 4w + 1) + \frac{2}{\eta^2}. \tag{34}$$

By a similar argument to the previous case, the Bayes optimal estimator and the corresponding Bayes risk is:

$$\hat{\mu}_{(\epsilon',\epsilon')} = \frac{1}{4+\frac{12}{\epsilon'^2}}(X_1 + X_2) + Z, \quad Z \sim \text{Laplace}\left(\frac{\epsilon'}{4\epsilon'^2 + 12}\right), \tag{35}$$

$$r([\epsilon',\epsilon']) = \frac{1}{12}\left(1 - \frac{1}{2+\frac{6}{\epsilon'^2}}\right). \tag{36}$$

Finally letting $U(\boldsymbol{\epsilon}) = c_1 r(\boldsymbol{\epsilon}) + c_2$. Take $U(\mathbf{0}) = 0 \implies c_1 = -12c_2$. And $\max_{\boldsymbol{\epsilon}} U(\boldsymbol{\epsilon}) = 1 \implies c_1 = 24(1-c_2)$. Simplifying gives us our desired result:

$$\mathbf{U} = \begin{bmatrix} U([0,0]^T) & U([0,\epsilon']^T) \\ U([\epsilon',0]^T) & U([\epsilon',\epsilon']^T) \end{bmatrix} = \begin{bmatrix} 0 & 2\left(3+\frac{24}{(\epsilon')^2}\right)^{-1} \\ 2\left(3+\frac{24}{(\epsilon')^2}\right)^{-1} & \left(1+\frac{3}{(\epsilon')^2}\right)^{-1} \end{bmatrix} \tag{37}$$

$\square$

## B.2 Proof of Theorem 1 and Theorem 2

We will begin with the proof of Theorem 2, which is standard and follows the typical proof of the Shapley value. We begin by proving $\phi_i(\boldsymbol{\rho})$ as defined in equation 7 satisfies axioms (B.i-iii). First assume $U(\boldsymbol{\rho}_{S\cup\{i\}}) = U(\boldsymbol{\rho}_{S\cup\{j\}}) \ \forall S \subset [N]\backslash\{i,j\}$, then:

$$\phi_i(\boldsymbol{\rho}) = \frac{\alpha(\boldsymbol{\rho})}{N} \sum_{S\subseteq[N]\backslash\{i\}} \frac{U(\boldsymbol{\rho}_{S\cup\{i\}}) - U(\boldsymbol{\rho}_S)}{\binom{N-1}{|S|}} \tag{38}$$

$$= \frac{\alpha(\boldsymbol{\rho})}{N}\left(\sum_{S\subseteq[N]\backslash\{i,j\}} \frac{U(\boldsymbol{\rho}_{S\cup\{i\}}) - U(\boldsymbol{\rho}_S)}{\binom{N-1}{|S|}} + \sum_{S\subseteq[N]\backslash\{i,j\}} \frac{\left(U(\boldsymbol{\rho}_{S\cup\{j\}\cup\{i\}}) - U(\boldsymbol{\rho}_{S\cup\{j\}})\right)}{\binom{N-1}{|S|+1}}\right) \tag{39}$$

$$= \frac{\alpha(\boldsymbol{\rho})}{N}\left(\sum_{S\subseteq[N]\backslash\{i,j\}} \frac{U(\boldsymbol{\rho}_{S\cup\{j\}}) - U(\boldsymbol{\rho}_S)}{\binom{N-1}{|S|}} + \sum_{S\subseteq[N]\backslash\{i,j\}} \frac{\left(U(\boldsymbol{\rho}_{S\cup\{i\}\cup\{j\}}) - U(\boldsymbol{\rho}_{S\cup\{i\}})\right)}{\binom{N-1}{|S|+1}}\right) \tag{40}$$

$$= \phi_j(\boldsymbol{\rho}), \tag{41}$$

proving axiom (B.i) is satisfied. For the proof that axiom (B.ii) is satisfied, we write:

$$\sum_i \phi_i(\boldsymbol{\rho}) = \frac{\alpha(\boldsymbol{\rho})}{N} \sum_i \sum_{S \subseteq [N]\setminus\{i\}} \frac{U(\boldsymbol{\rho}_{S\cup\{i\}}) - U(\boldsymbol{\rho}_S)}{\binom{N-1}{|S|}} \tag{42}$$

$$= \frac{\alpha(\boldsymbol{\rho})}{N} \left( \sum_i \sum_{S \subseteq [N]\setminus\{i\}} \frac{U(\boldsymbol{\rho}_{S\cup\{i\}})}{\binom{N-1}{|S|}} - \sum_i \sum_{S \subseteq [N]\setminus\{i\}} \frac{U(\boldsymbol{\rho}_S)}{\binom{N-1}{|S|}} \right) \tag{43}$$

$$= \alpha(\boldsymbol{\rho})U(\boldsymbol{\rho}) + \frac{\alpha(\boldsymbol{\rho})}{N} \left( \sum_i \sum_{\substack{S \subseteq [N]\setminus\{i\} \\ |S| < N-1}} \frac{U(\boldsymbol{\rho}_{S\cup\{i\}})}{\binom{N-1}{|S|}} - \sum_i \sum_{S \subseteq [N]\setminus\{i\}} \frac{U(\boldsymbol{\rho}_S)}{\binom{N-1}{|S|}} \right) \tag{44}$$

$$= \alpha(\boldsymbol{\rho})U(\boldsymbol{\rho}) + \frac{\alpha(\boldsymbol{\rho})}{N} \left( \sum_i \sum_{\substack{S \subseteq [N] \\ i \in S \\ |S| < N-1}} \frac{U(\boldsymbol{\rho}_S)}{\binom{N-1}{|S|-1}} - \sum_{\substack{S \subseteq [N] \\ |S| \leq N-1}} \frac{(N-|S|)U(\boldsymbol{\rho}_S)}{\binom{N-1}{|S|}} \right) \tag{45}$$

$$= \alpha(\boldsymbol{\rho})U(\boldsymbol{\rho}) + \frac{\alpha(\boldsymbol{\rho})}{N} \left( \sum_{\substack{S \subseteq [N] \\ |S| \leq N-1}} \frac{|S| U(\boldsymbol{\rho}_S)}{\binom{N-1}{|S|-1}} - \sum_{\substack{S \subseteq [N] \\ |S| \leq N-1}} \frac{(N-|S|)U(\boldsymbol{\rho}_S)}{\binom{N-1}{|S|}} \right) \tag{46}$$

$$= \alpha(\boldsymbol{\rho})U(\boldsymbol{\rho}), \tag{47}$$

thus proving axiom (B.ii) is satisfied. Finally, we note that (B.iii) is satisfied by linearity. Next, we establish the uniqueness of equation 7. To prove uniqueness, we take an approach that is standard in the literature where we define the unanimity game, show the uniqueness of the $\phi_i(\boldsymbol{\rho})$ in that case, and then argue that uniqueness follows from additivity (B.iii).

Define the unanimity utility, indexed by some $T \subseteq [N]$:

$$U_T(\boldsymbol{\rho}) = \begin{cases} 1 & \text{if } T \subseteq \text{supp}(\boldsymbol{\rho}) \\ 0 & \text{if } else. \end{cases} \tag{48}$$

$\{U_T\}_{T \subseteq [N]}$ form a linear basis for utility function such that any utility $U$ can be represented uniquely by a set of values $\{b_T\}_{T \subseteq [N]}$. In addition, by direct application of the axioms, it is easy to see that for the unanimity utility, the fair allocation $\phi_i^{(T)}(\boldsymbol{\rho})$ is unique and is of the form:

$$\phi_i^{(T)}(\boldsymbol{\rho}) = \begin{cases} \frac{\alpha(\boldsymbol{\rho})}{T} & \text{if } i \in T \\ 0 & \text{if } else. \end{cases} \tag{49}$$

Thus, for any utility $U$, the fair value is represented uniquely by $\sum_{T \subseteq [N]} b_T \phi_i^{(T)}(\boldsymbol{\rho})$, since this value is unique, it must be equivalent to equation 7.

Now we consider the proof of Theorem 1. By a similar argument to the above, we can establish that:

$$\phi_p(z, \boldsymbol{\rho}) = \frac{1}{N+1} \sum_{S \subseteq [N]} \frac{U(z, \boldsymbol{\rho}_S) - U(0, \boldsymbol{\rho}_S)}{\binom{N}{|S|}} \tag{50}$$

as well as:

$$\phi_i(z, \boldsymbol{\rho}) = \frac{1}{N+1} \sum_{\substack{S \subseteq [N]\setminus\{i\} \\ z' \in \{0,z\}}} \frac{1}{\binom{N}{|S|+\mathbb{1}(z'=1)}} \left( U(z', \boldsymbol{\rho}_{S\cup\{i\}}) - U(z', \boldsymbol{\rho}_S) \right) \tag{51}$$

$$\tag{52}$$

Applying the definition $U(0, \boldsymbol{\rho}) = 0$ we have

$$\phi_p(z, \boldsymbol{\rho}) = \frac{1}{N+1} \sum_{S \subseteq [N]} \frac{U(z, \boldsymbol{\rho}_S)}{\binom{N}{|S|}} \tag{53}$$

$$\phi_i(z, \boldsymbol{\rho}) = \frac{1}{N+1} \sum_{S \subseteq [N] \setminus \{i\}} \frac{1}{\binom{N}{|S|+1}} \left( U(z, \boldsymbol{\rho}_{S \cup \{i\}}) - U(z, \boldsymbol{\rho}_S) \right), \tag{54}$$

completing the proof.

### B.3  Error Computation for Section 4

In this section we prove Proposition 4 and 5 from which exact error expressions follow.

**Proposition 4.** *For the federated mean estimation problem described in Section 4, the expected mean-squared error is given by:*

$$\mathbb{E}\left[ \left( \hat{\theta}_i^p - \theta_i \right)^2 \right] =$$

$$r^2 \left( \sum_{j=1}^{N_2} w_{ij}^2 \cdot \frac{1}{n_j} + \frac{1}{N_1} w_{i0}^2 \frac{1}{\bar{n}} \right) + s^2 \left( \sum_{\substack{j=1 \\ j \neq i}}^{N_2} w_{ij}^2 + \frac{1}{N_1^2} \sum_{\substack{j=N_2+1 \\ j \neq i}}^{N_2+N_1} w_{i0}^2 + \left( \sum_{\substack{j=1 \\ j \neq i}}^{N_2} w_{ij} + \frac{1}{N_1} \sum_{\substack{j=N_2+1 \\ j \neq i}}^{N_2+N_1} w_{i0} \right)^2 \right), \tag{55}$$

*where $\bar{n} = \left( \frac{1}{N_1} \sum_{j=N_2+1}^{N_1+N_2} \frac{1}{n_j} \right)^{-1}$.*

*Proof.* Consider an estimator of the form $\hat{\theta}_i^p = \sum_{i=1}^{N} v_{ij} \hat{\theta}_j$, where user $j$ has $n$ samples, and $\theta_j$ is the local model of user $j$. By Theorem 4.2 of Donahue & Kleinberg (2021), the error can be written as:

$$\mathbb{E}\left[ \left( \hat{\theta}_i^p - \theta_i \right)^2 \right] = r^2 \sum_{j=1}^{N} v_{ij}^2 \cdot \frac{1}{n_j} + s^2 \left( \sum_{j \neq i} v_{ij}^2 + \left( \sum_{j \neq i} v_{ij} \right)^2 \right) \tag{56}$$

For $j = 1, \dots, N_2$, we have $v_{ij} = w_{ij}$. For $j = N_2 + 1, \dots, N_2 + N_1$, we have $v_{ij} = \frac{w_{i0}}{N_1}$. Finally, for $j > N_1 + N_2$, we have $v_{ij} = 0$. Thus the first term can be written as:

$$r^2 \sum_{j=1}^{N} v_{ij}^2 \cdot \frac{1}{n_j} = r^2 \left( \sum_{j=1}^{N_2} w_{ij}^2 \frac{1}{n_j} + \sum_{j=N_2+1}^{N_2+N_1} \frac{1}{n_j} \left( \frac{w_{i0}}{N_1} \right)^2 \right) \tag{57}$$

$$= r^2 \left( \sum_{j=1}^{N_2} w_{ij}^2 \frac{1}{n_j} + \frac{1}{N_1} w_{i0}^2 \frac{1}{\bar{n}} \right). \tag{58}$$

Making these same substitutions to $\sum_{j \neq i} v_{ij}^2$ and $\sum_{j \neq i} v_{ij}$ yields the desired result. $\square$

**Proposition 5.** *The error expression equation 55 is minimized if $\rho_i = 0$ with weights:*

$$w_{i0} = \frac{N_1}{N_1 + N_2 \frac{V_0}{V}}, \quad w_{ij} = \frac{V_0/V_j}{N_1 + N_2 \frac{V_0}{V}}. \tag{59}$$

*If $\rho_i = 1$ equation 55 is minimized by:*

$$w_{i0} = \frac{N_1}{N_1 + N_2 \frac{V_0}{V}} + \frac{N_2}{N_1 + N_2 \frac{V_0}{V}} \frac{s^2}{\bar{V}}, \tag{60}$$

$$w_{ij} = \frac{V_0/V_j}{N_1 + N_2 \frac{V_0}{\bar{V}}} - \frac{1}{N_1 + N_2 \frac{V_0}{\bar{V}}} \frac{s^2}{V_j}. \tag{61}$$

*Finally, if $\rho_i = 2$, equation 55 is minimized by:*

$$w_{i0} = \frac{N_1}{N_1 + N_2 \frac{V_0}{\bar{V}}} - \frac{N_1}{N_1 + N_2 \frac{V_0}{\bar{V}}} \frac{s^2}{V_i}, \tag{62}$$

$$w_{ij} = \frac{V_0/V_j}{N_1 + N_2 \frac{V_0}{\bar{V}}} - \frac{V_0/V_j}{N_1 + N_2 \frac{V_0}{\bar{V}}} \frac{s^2}{V_i} \tag{63}$$

$$w_{ii} = \frac{V_0/V_i}{N_1 + N_2 \frac{V_0}{\bar{V}}} + \frac{N_1 + N_2 \frac{V_0}{\bar{V}} - \frac{V_0}{V_i}}{N_1 + N_2 \frac{V_0}{\bar{V}}} \frac{s^2}{V_i} \tag{64}$$

*Proof.* First we will consider the case where $\rho_i = 1$. Considering the point where the derivative of equation 55 with respect to $w_{ik}, k \geq 1$ is equal to zero gives:

$$\frac{2r^2}{n_k} w_{ik} - \frac{2r^2}{\bar{n} N_1} \left( 1 - \sum_{j=1}^{N_2} w_{ij} \right) + s^2 \left( 2w_{ik} - 2\frac{N_1 - 1}{N_1^2} \left( 1 - \sum_{j=1}^{N_2} w_{ij} \right) + \frac{2}{N_1^2} \left( N_1 - 1 + \sum_{j=1}^{N_2} w_{ij} \right) \right) = 0, \tag{65}$$

$$\left( \frac{r^2}{n_k} + s^2 \right) w_{ik} = \left( \frac{r^2}{\bar{n}} + s^2 \right) \frac{w_{i0}}{N_1} - \frac{s^2}{N_1}. \tag{66}$$

It is easily verified from the second derivative that solving this equation gives us the unique minimum of equation 55. For ease of notation, define $V_k := \left( \frac{r^2}{n_k} + s^2 \right)$ and $V_0 := \left( \frac{r^2}{\bar{n}} + s^2 \right)$, $\bar{V} = \left( \frac{1}{N_2} \sum_{k=1}^{N_2} \frac{1}{V_k} \right)^{-1}$. Thus, we have:

$$w_{ik} = \frac{V_0 \frac{w_{i0}}{N_1} - \frac{s^2}{N_1}}{V_k}. \tag{67}$$

Noting that $w_{i0} + \sum_{j=1}^{N_2} w_{ij} = 1$, we have:

$$w_{i0} + \frac{N_2}{N_1} \frac{V_0}{\bar{V}} w_{i0} - \frac{N_2}{N_1} \frac{s^2}{\bar{V}} = 1, \tag{68}$$

$$w_{i0} = \frac{N_1}{N_1 + N_2 \frac{V_0}{\bar{V}}} + \frac{N_2}{N_1 + N_2 \frac{V_0}{\bar{V}}} \frac{s^2}{\bar{V}}, \tag{69}$$

$$w_{ij} = \frac{V_0/V_j}{N_1 + N_2 \frac{V_0}{\bar{V}}} - \frac{1}{N_1 + N_2 \frac{V_0}{\bar{V}}} \frac{s^2}{V_j}. \tag{70}$$

This completes the proof for those users $i$ such that $\rho_i = 1$. When $\rho_i = 2$, the gradient condition with respect to $k \geq 1$, $k \neq i$ is:

$$w_{ik} V_k = \frac{V_0}{N_1} w_{i0}, \tag{71}$$

and similarly, the gradient condition when $k = i$ is:

$$w_{ii} V_i + w_{i0} \frac{N_2 V_0}{N_1 \bar{V}} + \frac{s^2}{V_i} = 1. \tag{72}$$

Combining these together gives our desired result. $\rho_i = 0$ $\qquad \square$

## B.4 Proof of Theorem 3

The symmetric Nash equilibria of our game is characterized Cheng et al. (2004) by the minimizers of

$$\min_p \sum_{s \in \mathcal{E}} [u(s,p) - u(p,p)]_+^2, \tag{73}$$

where $u(s,p)$ is the utility a user when they choose privacy level $\rho_i = s$, and all other users play mixed strategy $\mathbf{p}$, and $u(p,p) = \mathbb{E}_{s \sim \mathbf{p}}[u(s,p)]$. Since our action space is binary, there are only two terms in this sum. Applying the definition of $u$ and writing out both terms of this sum yields:

$$\sum_{s \in \mathcal{E}} [u(s,p) - u(p,p)]_+^2 = [u(\rho_1,p) - u(p,p)]_+^2 + [u(\rho_2,p) - u(p,p)]_+^2 \tag{74}$$

$$= [c(1-p) - \alpha(\phi(p,p) - \phi(\rho_1,p))]_+^2 + [c(1-p) - \alpha(\phi(p,p) - \phi(\rho_2,p))]_+^2 \tag{75}$$

$$= [(1-p)(c - \alpha\gamma(p))]_+^2 + [-p(c - \alpha\gamma(p))]_+^2, \tag{76}$$

where we define $\gamma(p) := \phi(\rho_2,p) - \phi(\rho_1,p)$. $\gamma$ is an important quantity in this problem that described the relative increase in payment a user receives for choosing a higher privacy level when the other users choose mixed strategy $\mathbf{p}$. In general, to say something about the equilibria, we must say something about $\gamma$. We can now use Assumptions 1 and 2, as well as the definition of $\phi(\cdot;\cdot)$ to establish properties of $\gamma$. First we show $\gamma(p) \geq 0$ using monotonicity of $U$:

$$\gamma(p) = \phi(\rho_2,p) - \phi(\rho_1,p), \tag{77}$$

$$= \mathbb{E}_{\substack{\rho_j \sim \mathbf{p} \\ \rho_i = \rho_2'}} \left[ \frac{1}{N} \sum_{S \subseteq [N] \setminus \{i\}} \frac{1}{\binom{N-1}{|S|}} \left( U(\boldsymbol{\rho}_{S \cup \{i\}}) - U(\boldsymbol{\rho}_S) \right) \right]$$

$$- \mathbb{E}_{\substack{\rho_j \sim \mathbf{p} \\ \rho_i = \rho_1'}} \left[ \frac{1}{N} \sum_{S \subseteq [N] \setminus \{i\}} \frac{1}{\binom{N-1}{|S|}} \left( U(\boldsymbol{\rho}_{S \cup \{i\}}) - U(\boldsymbol{\rho}_S) \right) \right], \tag{78}$$

$$= \frac{1}{N} \sum_{S \subseteq [N] \setminus \{i\}} \frac{1}{\binom{N-1}{|S|}} \mathbb{E}_{\substack{\rho_j \sim \mathbf{p} \\ j \neq i}} \left[ U(\boldsymbol{\rho}_{S \cup \{i\}}^{(i+)}) - U(\boldsymbol{\rho}_{S \cup \{i\}}^{(i-)}) \right] \geq 0. \tag{79}$$

In equation 78 we have used the definition of the fair value from Theorem 2, and in equation 79, we have simplified the expression, exchanged the sum and expectation, and used the fact that the expectation of a non-negative random variable is non-negative.

Next, we will show that under Assumption 2 (and our assumption of equal marginal contribution) we also have $\gamma'(p) \geq 0$. Assume $p_2 > p_1$, and let $b(n,p) = \binom{N}{n} p^i (1-p)^{N-i}$:

$$\gamma(p_2) - \gamma(p_1) = \frac{1}{N} \sum_{S \subseteq [N] \setminus \{i\}} \frac{1}{\binom{N-1}{|S|}} \left( \mathbb{E}_{\substack{\rho_j \sim \mathbf{p}_2 \\ j \neq i}} \left[ U(\boldsymbol{\rho}_{S \cup \{i\}}^{(i+)}) - U(\boldsymbol{\rho}_{S \cup \{i\}}^{(i-)}) \right] - \mathbb{E}_{\substack{\rho_j \sim \mathbf{p}_1 \\ j \neq i}} \left[ U(\boldsymbol{\rho}_{S \cup \{i\}}^{(i+)}) - U(\boldsymbol{\rho}_{S \cup \{i\}}^{(i-)}) \right] \right) \tag{80}$$

$$= \frac{1}{N} \sum_{S \subseteq [N] \setminus \{i\}} \frac{1}{\binom{N-1}{|S|}} \sum_{n=0}^{N} (b(n,p_2) - b(n,p_1)) \Delta_i U(\boldsymbol{\rho}(n)) \quad \text{s.t. } n_{private}(\boldsymbol{\rho}(n)) = N - n \tag{81}$$

Now note that $b(n,p_2) - b(n,p_1)$ is zero-mean, and decreasing, furthermore, $\Delta_i U(\boldsymbol{\rho}(n))$ is non-negative and non-increasing. Let $n^*$ represent the smallest value of $n$ such that $b(n,p_2) - b(n,p_1)$ is negative. Then we have:

$$\Delta_i U(\boldsymbol{\rho}(n)) = \sum_{n=0}^{n^*-1} (b(n,p_2) - b(n,p_1)) \Delta_i U(\boldsymbol{\rho}(n)) + \sum_{n=n^*}^{N} (b(n,p_2) - b(n,p_1)) \Delta_i U(\boldsymbol{\rho}(n)) \tag{82}$$

$$\geq \left( \sum_{n=0}^{n^*-1} b(n,p_2) - b(n,p_1) \right) (\Delta_i U(\boldsymbol{\rho}(n^*-1)) - \Delta_i U(\boldsymbol{\rho}(n^*))) \tag{83}$$

$$\geq 0. \tag{84}$$

With the knowledge that $\gamma(p) \geq 0$ and $\gamma'(p) \geq 0$ we can compute $p^*$ for three distinct cases. Defining $\gamma_{max} := \max_p \gamma(p)$ and $\gamma_{min} := \min_p \gamma(p)$, we have:

**Case 1** $c - \alpha\gamma_{max} > 0$:

$$\sum_{s \in \mathcal{E}} [u(s,p) - u(p,p)]_+^2 = [(1-p)(c - \alpha\gamma(p))]_+^2 \tag{85}$$

Since this quantity is non-negative, it is clearly minimized when $p^* = 1$, where it is exactly 0. Furthermore, since $c - \alpha\gamma_{max} > 0$ is satisfied with strict inequality, it is the unique minimizer.

**Case 2** $c/\alpha \in [\gamma_{min}, \gamma_{max}]$:

$$\sum_{s \in \mathcal{E}} [u(s,p) - u(p,p)]_+^2 = [(1-p)(c - \alpha\gamma(p))]_+^2 + [-p(c - \alpha\gamma(p))]_+^2 , \tag{86}$$

In the above case, this is minimized when $p^* \in \gamma^{-1}(c/\alpha)$.

**Case 3** $c - \alpha\gamma_{min} < 0$:

$$\sum_{s \in \mathcal{E}} [u(s,p) - u(p,p)]_+^2 = [-p(c - \alpha\gamma(p))]_+^2 , \tag{87}$$

In the above case, the expression is minimized when $p^* = 0$. To summarize, we have:

$$p^*(\alpha) = \begin{cases} 1 & \text{if } \alpha < \frac{c}{\gamma_{max}} \\ \gamma^{-1}(c/\alpha) & \text{if } \alpha \in [\frac{c}{\gamma_{max}}, \frac{c}{\gamma_{min}}] \\ 0 & \text{if } \alpha > \frac{c}{\gamma_{min}} \end{cases} . \tag{88}$$

This establishes that the Nash equilibrium is cleanly separated into three regions. From this fact, we are able to show that the optimal strategy of the platform is also separated into three regions. We consider a platform that solves the following problem, where we define $U(p) := \mathbb{E}_{\rho_i \sim \mathbf{p}}[U(\boldsymbol{\rho})]$:

$$\min_\alpha (1 - \alpha)U(p^*(\alpha)), \tag{89}$$

Clearly, when privacy sensitivity is large, specifically, when $c \geq \gamma_{max}$ then $\alpha^* = 0$ is the optimal solution, since $p^*(\alpha) = 1$ for all $\alpha < 1$, and for $\alpha > 1$ the objective becomes negative.

Alternatively, when $c$ is very small, we can determine the optimal value as follows. We first note that Assumption 1 implies that $U(p)$ is a decreasing function of $p$. Thus the condition for $\alpha^* = \frac{c}{\gamma_{min}}$ is:

$$\frac{1 - c/\gamma_{min}}{1 - \alpha} > \frac{U(p^*(\alpha))}{U(0)} \quad \forall \alpha < c/\gamma_{min}. \tag{90}$$

Since the left-hand side takes value $\frac{1}{1-\alpha}$ at $c = 0$, while the right-hand side is 1, as well as the fact that both sides are continuous, by the Intermediate Value Theorem, (and our previous result, which implies that for $c$ large enough this condition does not hold), there is some minimum $c_{th}$, where this condition fails. Thus we conclude, there are three regions:

(1) a region where $c \leq c_{th}$ is small, and $\alpha^*$ is the smallest $\alpha$ such that $p^* = 0$, (2) an intermediate region where a symmetric mixed strategy is played, and (3) a region where $c \geq \gamma_{max}$ , and $\alpha^* = 0, p^* = 1$

### B.4.1 Exact Calculation for Example

In this section, we work thought our example in Section 5, showing that using Theorem 3 and some basic calculus, we can determine valuable information about $\alpha^*$ The utility function is of the form:

$$\mathbf{U} = \begin{bmatrix} 0 & 2/3 \\ 2/3 & 1 \end{bmatrix} . \tag{91}$$

$$\mathbf{\Phi}_1^{(2)} = \alpha \begin{bmatrix} 0 & 2/3 \\ 0 & 1/2 \end{bmatrix}, \quad \mathbf{\Phi}_2^{(2)} = \alpha \begin{bmatrix} 0 & 0 \\ 2/3 & 1/2 \end{bmatrix} \tag{92}$$

When users play a mixed strategy $p$, the utility $U(p)$ can be written as

$$U(p) = -\frac{1}{3}p^2 - \frac{2}{3}p + 1. \tag{93}$$

The gamma function likewise can be computed as

$$\gamma(p) = \frac{1}{2} + \frac{1}{6}p. \tag{94}$$

Thus, we find that $\gamma_{max} = \frac{2}{3}$ and $\gamma_{min} = \frac{1}{2}$. This allows us to compute $p^*(\alpha)$ :

$$p^*(\alpha) = \begin{cases} 1 & \text{if } \alpha < \frac{3c}{2} \\ 6\frac{c}{\alpha} - 3 & \text{if } \frac{3c}{2} \le \alpha \le 2c \\ 0 & \text{if } \alpha > 2c \end{cases}. \tag{95}$$

From this and Theorem 3, we immediately know $\alpha^* = 0$ when $c \ge \frac{2}{3}$. Next, we can determine $c_{th}$. First, we compute $U(p^*(\alpha))$:

$$U(p^*(\alpha)) = \begin{cases} 0 & \text{if } \alpha < \frac{3c}{2} \\ \frac{8c}{\alpha} - \frac{12c^2}{\alpha^2} & \text{if } \frac{3c}{2} \le \alpha \le 2c \\ 1 & \text{if } \alpha > 2c \end{cases}. \tag{96}$$

The threshold is concerned only with $\alpha \le 2c$. We note that for $c < \frac{1}{3}$, the function $(1-\alpha)U(p^*(\alpha))$ is monotone on $0 \le \alpha \le 2c$, and attains the value $1 - 2c$ at $\alpha = 2c$. However, for $c < \frac{1}{3}$, it exceeds $1 - 2c$ at it's maximum value at $\alpha^* = \frac{6c}{3c+2}$. Thus, $c_{th}$ is $\frac{1}{3}$. To summarize, the optimal $\alpha^*$ is:

$$\alpha^* = \begin{cases} 2c & \text{if } c < \frac{1}{3} \\ \frac{6c}{3c+2} & \text{if } \frac{1}{3} \le c \le \frac{2}{3} \\ 0 & \text{if } c > \frac{2}{3} \end{cases}, \tag{97}$$

and making the neccessary substitutions generates the plots in Fig 8.

## C  Monotonicity of Utility

When beginning this work, the dearth of algorithms that supported heterogeneous privacy constraints surprised us, given the increasing number of privacy options available to users. All of the algorithms that did exist were provably sub-optimal Hu & Gong (2020), or placed constraints on privacy parameters to prove approximate optimality Fallah et al. (2022). In both of these works, the pathology of the algorithm leads to error that is not monotonically decreasing in $\boldsymbol{\rho}$. For DP-based notions of privacy, which both of the aforementioned works are, one can prove that an optimal error must be monotonic. This observation inspired a recent work that studies a *saturation* phenomenon Chaudhuri & Courtade (2023). Similar ideas can also be found in Cummings et al. (2023). The idea is that an optimal algorithm will sometimes give users that choose a large $\epsilon_i$ more privacy than they asked for, to ensure that it still efficiently uses information from users $j$ with $\epsilon_j \ll \epsilon_i$.

## D  Solving the Federated Mean Estimation Mechanism Design Problem

In this section we discuss how we produce Fig 7. Algorithm 1 describes the process. In the first step, we exploit the symmetry of the utility function by exploiting the fact that the utility does not change when $\rho_i$ is exchanges between two users with the same $n_i$. This greatly reduces the number of payment functions that need to be calculated. Next, we compute the payment functions, again exploiting symmetry whenever possible to reduce calculations. Finally, for each partition, we efficiently search through the partition to see if there is a $\boldsymbol{\rho}$ that leads to a NE.

---

**Algorithm 1:** Find optimal $\alpha$

---

**input** : $n_i$, $c_i$ : $i = 1 \ldots, N$
**output:** $\alpha^*$
nArray($i$) $\leftarrow n_i$, $i = 1 \ldots, N$;
cArray($i$) $\leftarrow c_i$, $i = 1 \ldots, N$;
partitions $\leftarrow$ GetValidPartitions($n_i : i = 1 \ldots, N$) ;     /* all $\boldsymbol{\rho}$ that produce unique $U$ */
**for** $i = 1$ **to** len(partitions) **do**
   $\boldsymbol{\rho} \leftarrow$ partitions($i$) ;     /* one representative $\boldsymbol{\rho}$ from the partition */
   **for** $j = 1$ **to** $N$ **do**
      | phi($j$) $\leftarrow$ Shapley($\boldsymbol{\rho}$, $j$, nArray) ;     /* Actual code skips repeated calculations */
   **end**
   **for** $\alpha \in$ grid **do**
      neExists $\leftarrow$ TreeSearch($\alpha \times$ phi, cArray) ;  /* Check if any $\boldsymbol{\rho}$ in the partition is NE */
      **if** neExists **then**
         currUtil $\leftarrow (1 - \alpha) \times$ Utility ($\boldsymbol{\rho}$, nArray);
         **if** currUtil $>$ maxUtil **then**
            | $\alpha^* \leftarrow \alpha$ ;     /* Update $\alpha^*$ if needed */
            | maxUtil $\leftarrow$ currUtil;
         **end**
      **end**
   **end**
**end**

---

**Comment on Fig 7** The observant reader will notice there is almost always one user with $\rho_i = 1$. This is an artifact of our model, where if only one user chooses $\rho_i = 1$, it essentially behaves the same from a utility perspective as if it has chosen $\rho_i = 2$, but gets a reduced privacy sensitivity. It is possible to show that this means that other than the all zeros NE, there will always be one user with $\rho_i = 1$.

## E   Equilibria for Binary Privacy Level with Two Different Privacy Sensitivities

Let $\mathbf{p} = [p \; (1 - p)]^T$ be the mixed strategy of user 1 and let $\mathbf{q} = [q \; (1 - q)]^T$ be the mixed strategy of user 2. When they play these respective strategies, the utility of user 1 is:

$$
\begin{aligned}
u_1(\mathbf{p}, \mathbf{q}) &= pq\alpha\phi(\rho'_1, \rho'_2) + p(1 - q)\alpha\phi(\rho'_1, \rho'_2) + (1 - p)q\alpha\phi(\rho'_2, \rho'_1) + (1 - p)(1 - q)\alpha\phi(\rho'_2, \rho'_2) + -c_1(1 - p) \\
&= p\alpha\phi(\rho'_1, \mathbf{q}) + (1 - p)\alpha\phi(\rho'_2, \mathbf{q}) - c_1(1 - p) \\
&= p\left(c_1 - \alpha\gamma(q)\right) + \alpha\phi(\rho'_2, \mathbf{q}) - c_1.
\end{aligned}
$$

By a symmetric argument, we also have that

$$u_2(\mathbf{p}, \mathbf{q}) = q\left(c_2 - \alpha\gamma(p)\right) + \alpha\phi(\rho'_2, \mathbf{p}) - c_2. \tag{98}$$

We are interested in characterizing the best response maps:

$$\mathrm{BR}_1(\mathbf{q}; \alpha) = \arg\max_{\mathbf{p}} u_1(\mathbf{p}, \mathbf{q}) \quad \mathrm{BR}_2(\mathbf{p}; \alpha) = \arg\max_{\mathbf{q}} u_2(\mathbf{p}, \mathbf{q}), \tag{99}$$

since their intersection characterize the set of NEs.

We begin with finding an analytic expression for $\mathrm{BR}_1(\mathbf{q}; \alpha)$, which, we will break into three distinct cases:

**Case 1:**  $c_1 - \alpha\gamma_{max} > 0$

In this case, the constant factor in front of $p$ is always positive (invoking the monotonicity and non-negativity we proved in the previous section under the assumptions), thus the best response is:

$$\mathrm{BR}_1(\mathbf{q}; \alpha) = [1 \; 0]^T \; \forall \alpha < \frac{c_1}{\gamma_{max}}. \tag{100}$$

**Case 2:** $c_1 - \alpha\gamma_{min} < 0$

In this case, by a similar argument to before, the constant factor in front of $p$ is always negative, thus the best response is:

$$\text{BR}_1(\mathbf{q};\alpha) = [0\ 1]^T \ \forall \alpha > \frac{c_1}{\gamma_{min}}. \tag{101}$$

**Case 3:** $\alpha \in \left[\frac{c_1}{\gamma_{max}}, \frac{c_1}{\gamma_{min}}\right]$

In this case, the sign of the factor in front of $p$ changes with $\mathbf{q}$. We can write the best response piece-wise as:

$$\text{BR}_1(\mathbf{q};\alpha) = \begin{cases} [1\ 0]^T & \text{if } c_1 - \alpha\gamma(q) > 0 \\ \{[a\ b]^T : a,b \geq 0,\ a+b=1\} & \text{if } c_1 - \alpha\gamma(q) = 0 \\ [0\ 1]^T & \text{if } c_1 - \alpha\gamma(q) < 0 \end{cases} \tag{102}$$

This same analysis can be applied to $\text{BR}_2(\mathbf{p};\alpha)$. The NE is characterized by the sets where these two maps intersect. The following table summarize the equilibria $p^*, q^*$, written as scalars for readability.

| | $\alpha \leq \frac{c_1}{\gamma_{max}}$ | $\alpha \in \left[\frac{c_1}{\gamma_{max}}, \frac{c_1}{\gamma_{min}}\right]$ | $\alpha > \frac{c_1}{\gamma_{min}}$ |
|---|---|---|---|
| $\alpha \leq \frac{c_2}{\gamma_{max}}$ | $(1,1)$ | $(0,1)$ | $(0,1)$ |
| $\alpha \in \left[\frac{c_2}{\gamma_{max}}, \frac{c_2}{\gamma_{min}}\right]$ | $(1,0)$ | $\left\{(1,0),(0,1),\left(\gamma^{-1}\left(\frac{c_1}{\alpha}\right),\gamma\left(\frac{c_2}{\alpha}\right)\right)\right\}$ | $(0,1)$ |
| $\alpha > \frac{c_2}{\gamma_{min}}$ | $(1,0)$ | $(1,0)$ | $(0,0)$ |

When $\alpha$ is below the threshold for the two users (the top left entry), both $c_1$ and $c_2$ are too small for it to be worthwhile for the users to participate at the lower privacy option. Conversely, if $\alpha$ is above the threshold for both users, then both users choose the less private option. When neither of these extremes occur the results are more nuanced.

