# OpenReview forum: "The Fair Value of Data Under Heterogeneous Privacy Constraints in Federated Learning"
_TMLR — Accepted by TMLR_

### Review · Reviewer_u8qr · 2023-08-21

**Summary Of Contributions:**

This paper studies the fair payment allocation problem in federated learning based on Shapley values. It builds some game-theoretic framework, defines the utility functions for both the platform (server) and the users, and demonstrates some properties of the framework on a mean estimation problem.

**Audience:**

Yes

**Broader Impact Concerns:**

None.

**Claims And Evidence:**

Yes

**Requested Changes:**

* Improve presentation

* Dive deeper into the implications of the theorems.

* Provide a formal theorem with different c_i's.

* Discuss how do solve the fair allocation problem (19).

**Strengths And Weaknesses:**

Strengths:

* The problem is timely and interesting.
* The mean estimation example can demonstrate how the utility changes when users have different privacy requirements.
* Defining utility functions based on the Shapley value is meaningful.

Weaknesses:

* Presentation needs to be largely improved. Some terminology is not consistent with prior literature, and notations, equations, etc throughout the paper are not self-contained.

-- The definition of \epsilon-privacy algorithm is irrelevant to the \epsilon-DP notion in the DP literature. I suggest to name it differently to void confusion.

-- When epsilon=2, do the users send models or data? Regarding this, seems the text in Page 1-3 is inconsistent with Figure 2.

-- The epsilons defined in Def 1-2 is based on DP, which is different from the implications of epsilons in {0,1,2} described before. Also, why do we need coordinate-wise DP (Definition 1)? What is the definition of coordinates? Does epsilon_i correspond to user i or the i-th gradient/model dimension?

-- The sensitivity function is not defined. c_i is the sensitivity of what function defined over user data i? Why is it a function of epsilon_i? In my understanding, the sensitivity defined in privacy literature is not dependent on the privacy parameters, but rather, a property of the function and the data. As a result, I cannot parse the formulation of Eq. (2). When I read towards the very end of the paper, I realized (may be wrong) that c_i may refer to how much a user is sensitive to privacy protection, as opposed to the sensitivity where we usually need to bound to guarantee DP.

-- Is Section 4.1 is plotting Eq (12) or (17), where (12) is the payment function and (17) is the utility function? The first sentence of Section 4.1.1 points out that it plots ‘how much each user’s utility differs from U(ε)/N’; however, the following text (and the y label) seems to infer that the y-axis is the payment each client receives. Eq. (12) is more expensive to compute, but more interesting.


* The implications of the presented theorems are limited. Theorem 1 & 2 are listing the standard results of Shapley value. Example 1 gives a concrete example on a toy problem where we can calculate the utility values. What is the insight from this example on the coalition between the platform and all users? The fair payment depends on the (1) Shapley value calculation, and (2) the definition of the utility function. In the mean estimation case, (1) can be expensive and (2) can be defined and estimated in a straightforward way. The paper doesn’t discuss more general cases.

* In Section 5, is the ‘identical statistical marginal distribution’ assumption reasonable? Why is the platform utility function in Eq (19) in this section different from the one discussed in (3)? Additional clarification is needed. Theorem 3 also assumes that all clients share the same privacy parameter c. While Section 5.2 studies the case where users have different privacy tolerance levels, it would be better to provide a similar theorem where each user i has a different c_i.

* The mechanism design part formalizes the optimization problem (which is difficult to solve), characterizes some properties of the solutions under restrictive assumptions, and visualizes the solutions on a mean estimation problem; but doesn’t provide enough insights on meaningful algorithms to find the solution.

---

### Review · Reviewer_LbcM · 2023-08-28

**Summary Of Contributions:**

The paper models users in a federated learning environment as a coalition game. Where users can choose different privacy levels to share their data and they will get paid based on their choices and the decision of the platform. Essentially if the platform decides to join the coalition then each client will get paid. This will allow platforms to simulate how different payment strategies can affect the decision making of the users and if that payment strategy will result in a stable coalition. Finally the authors evaluated a few simple settings to show effectiveness of their approach.

**Audience:**

No

**Claims And Evidence:**

Yes

**Requested Changes:**

- Make it clear why this problem is coalition game rather than any other mechanism
- Complexity analysis of the solutions
- Motivations on practicality of the work

**Strengths And Weaknesses:**

Modeling users and federated learning platforms using game theory is logical, and much of the knowledge from game theory can be applied to this type of problem, similar to the work presented.

The paper chose to frame the issue within the context of a coalition game. The primary coalition concerns the platform's decision on whether or not to join with users; without this, the coalition has no value. It might simplify the problem if rather than the platform acting as a player deciding to join a coalition, this decision became a component of each player's utility, based on the size of the coalitions. For example, they have set requirements on the number of users with specific privacy choices in each coalition.  This would shift the problem to be more like a mechanism design, primarily aiming to design a good payoff mechanism  for users. While this might not significantly alter the results, it could improve the paper's readability.


The paper models the problem at a very abstract level and it is not clear to me if we found any solution with this level of abstraction that would reflect how real world implementations will look like. For example, the current modeling is that users will decide between a few privacy choices and then the platform decides if he wants to join or not. In practice do the users complete information about the platform utility function ? Does it make any difference?


The reward functions in the paper have exponential complexity, making it challenging to see their practicality in scenarios with a large number of users.


A minor comment that the paper uses epsilon in different parts of the paper to denote different meanings. In section 2.1 the authors use epsilon as in differential privacy epsilon but then later on they used epsilon to generally define privacy levels. The authors should try to use different notations or at least styles to improve readability.


One general comment: the interesting part of the paper is mechanism design and the game theory part of the work where it would be more appropriate in a game theory related venue.

---

### Review · Reviewer_EKtk · 2023-09-05

**Summary Of Contributions:**

The paper considers the problem of pricing data fairly in data acquisition and federated learning setting where data privacy is a major consideration.

**Audience:**

Yes

**Broader Impact Concerns:**

No broader impact concern; the goal here is privacy + fairness

**Claims And Evidence:**

Yes

**Requested Changes:**

To see the paper accepted, I would like to see the following:
1) Completely revamp the discussion of and connection with differential privacy in the paper to fix the issues mentioned above.
2) Improve the related work to provide a more exhaustive view of the work on data acquisition/mechanism design with privacy costs and/or under differential privacy
3) Clearly compare the results of this paper, especially Section 5, to previous work.

**Strengths And Weaknesses:**

Strengths:
- The interaction between fairness and privacy here is interesting
- The design space, which tries to consider the economics of privacy in a more general sense than differential privacy by extending the meaning of $\varepsilon$ is definitely a nice addition/novelty to the paper.

Weaknesses:
- A lot of the discussions of/connections with differential privacy are imprecise and/or erroneous. For example, it is known that sending the model directly to the platform without adding noise is not DP for any finite $\epsilon$. Yet, on page 4, the authors claims their $\epsilon = 2$ example is consistent to the standard definition of DP given in Definition 1.
- The related work is missing a lot of references. There has been a lot of work on mechanism design for data acquisition where the motivation is that agents incur privacy costs to release their data, and a lot of work trying to address this adding a differential privacy dimension. The current discussion only has a small subset of these papers. The statement that an economic theory for the value is data is still nascent with cites from 2019 also misses a lot of references in the computer science/EC literature starting 2013-2015.
- I worry about the novelty of Section 3. Theorems 1 and 2 seem to be standard Shapley value arguments/proofs.
- I also worry about the novelty of Section 5. The setting here seems very simple with only two possible levels of privacy, when a lot of previous work have considered non-binary and even continuous privacy sensitivities (e.g. Ghosh-Ligett 2013, Hsu et al. 2014, Cummings et al. 2015, Chen et al. 2018, Fallah et al. 2022, Cummings et al. 2023, many of which are not cited). The current paper seems behind the state of the art when it comes to mechanism design with privacy considerations.

---

### Review · Reviewer_RwJN · 2023-09-13

**Summary Of Contributions:**

Overview: The work studies the problem of a data aggregator attempting to learn from a collection of users who are heterogeneous in their privacy preferences. In this set-up, data subjects have multiple “privacy levels” at which they can choose to participate. For example, they may be able to participate with eps=1 or eps=2. The authors pose this as a mechanism design problem where the aggregator is required to come up with a payment function (monetary or otherwise) that incentivizes data subjects to participate with weaker privacy guarantees (e.g. a larger epsilon), while maximizing their profit (final utility - payments). The key contribution of the work is define what it means for such a payment structure to be “fair”. In the federated learning setting, a key component of the set of axioms for a “fair” payment structure set out by the authors is that two subjects should be compensated equally if their contribution to the utility is equal. A subject whose data does not impact the utility should not be compensated.

Given a utility function, the authors show at both a general level, and in specific examples, how to construct a “fair” payment structure. They give a specific, small scale example, to show how the optimal fair payment favors data subjects with highly influential (e.g. high quality) data, or who are willing to participate with lower levels of privacy. They also show how differential levels of heterogeneity in the data subjects can shift the relative payment between the subjects.

Comments:
Given the assumptions set forth in the paper, I think the formulation of the problem is nice. The axioms make sense. I thought the examples in Section 4 and Section 5.1 do a great job of illustrating how the optimal solutions behave. I found Figure 5 particularly interesting as it showed how heterogeneity in the user population plays a role in defining the optimal scheme.

My biggest concern with this work is the underlying assumption that a data subjects privacy preferences are a) a choice, and b) independent of their data. In practice, I think this is unlikely to be true. For example, data subjects with anomalous data (e.g. who are positive for a disease) are more likely to have a higher preference for privacy. Also, data subjects from historically discriminated against demographics are also more likely to place a higher value on privacy. I think this affects the work in a couple of distinct ways.
1. Given that the complex societal factors that go into privacy preferences, it seems worth examining whether paying data subjects based on their willingness to provide data with lower privacy protections can ever be “fair”. It seems like these schemes will always pay less to marginalized communities who may be more wary of participating and hence have a higher cost for participating.
2. Payment schemes like this bias the population whose data is collected. For example in Section 5.2, there are regimes where users with higher c (cost for participating) do not participate. If privacy preferences are correlated with data values, then this will create bias in the result. For example, entire populations may be missing from the sample because their data was too expensive to buy, exacerbating fairness concerns in the trained model.
3. In all the examples given in the paper, in order to find an optimal fair payment scheme, the aggregator needed to know the privacy preferences of each data subject. If this is correlated with the data, then I think this choice needs justification. I would like to see either a motivating example where privacy preferences being public can be justified, or some indication of whether this assumption could be avoided in the future.

This work is first step towards defining fairness in this setting so I don’t expect it to have answers to all these concerns. I don’t know of a solution that takes these factors into account and appropriately incentivizes participation. However, since the key contribution of this work is to set up a fairness definition, I would like to see these points discussed and more motivation for the choices that the authors have made.

I thought that the notation could be confused. Examples:
-  it was sometimes not obvious when something was a function that varied with data subjects or was a fixed universal constant (e.g. s and t in section 4 are originally defined in a way that makes this unclear).
- There is fluctuation in when p, \bf{p} and \bf{p}^* are used and I struggled to keep them straight. Why does Theorem 3 a used \bf{p}^* but b uses just \bf{p}? Should p be \bf{p} in eqn 21?
- \bf{epsilon} is a vector but is used in inequalities, how is this defined?
- In Figure 6, the labelling of platform profit by coloring in the region of the graph makes it seem like the profit is related to the area between the curves.
- It was not clear to me how Figure 7 compared the payments to user 1 and user 2? It just shows user 2’s payment, no?


Minor Comments:
- I would like to see the authors be a bit more careful about what they refer to as “privacy”. For example, in the introduction, the authors describe FL and just sending data to the server as distinct privacy levels. These are certainly not clearly providing different levels of privacy (in fact, we know in some settings that FL does not provide much additional privacy protections). I also think this highlights an interesting question about communicating privacy level choices to users (that is outside the scope of this paper) since FL sounds more private than it is (in my opinion).
- Typo: page 3 “but only places restriction only on relative”
- Figure 3, introduce the objective function outside the caption.
- Typo: Theorem 2 “Then for alpha(epsilon) satisfies”
- Introduce notation used in eqn 13
- After eqn 16, make it clear that Proposition 16 is in the appendix.
- Figure 5b, is there a typo in the labelling of the second and 4th columns?

**Audience:**

Yes

**Claims And Evidence:**

Yes

**Requested Changes:**

- Motivating example where privacy preferences can be public, or discussion of potential directions for eliminating this assumption.
- Discussion of how the correlation between a user's data and their privacy preferences affects the fairness of the payment scheme. In relation to both lower payments for more protective data subjects, and potential bias in the final result.

**Strengths And Weaknesses:**

See above

---

### Comment · Action_Editors · 2023-09-13
**Revision and subsequent discussion**

Dear authors, and reviewers,

The paper has now received 4 reviews from experts on different aspects of the work, including privacy, fairness, mechanism design, and federated learning. The reviewers have identified multiple concrete areas for revision and improvement and raised questions that include (but are not limited to) presentation, assumptions, related work, notations, results, and scalability of the proposal.

The authors have posted responses to the comments of three reviewers already but have not revised the paper as of yet. I suggest that the authors revise their paper based on the reviewers' extensive feedback in the next 2 weeks and post a revised version of the paper for a subsequent discussion with the reviewers.

There was also a concern around fit for TMLR by Reviewer LbcM. Given the broad scope of TMLR, and also the engaging reviews from the existing reviewer pool, I'd say that the paper is of interest to a non-trivial fraction of TMLR's audience, and may proceed to be considered for publication.

Thanks,\
Ahmad

---

### Decision · Action_Editor_hc53 · 2023-11-21

**Recommendation:** Accept with minor revision

**Comment:**

This paper proposes a framework based on Shapley value in game theory to understand the incentive structure for users to balance their privacy and utility, where the platform gives incentives to the users to choose to be less private. The framework adopts from Shapley value three axioms (fairness, efficiency, additivity), based on which derives theoretical results that characterize the optimal incentives that a platform should pay users for their data. The rest of the paper is dedicated to further developing these ideas through several examples.

The paper was reviewed by four expert reviewers on different aspects of the work with expertise ranging from federated learning, mechanism design, privacy, and fairness. The reviewers and authors engaged in a revision/response process and the reviewers have now submitted their recommendations on this paper. Overall, all reviewers and myself have found merit in the paper for publication. However, the reviewers have still identified several points that require another revision. In what follows, I describe the required revisions (based in part on the decision letter of the reviewers that was shared directly with the AE and EICs and in part on my own reading of the paper). The paper is recommended to be accepted subject to the authors incorporating the following revisions in their camera-ready version and providing a point-by-point response.

- The paper as it stands is hard to follow because the examples keep changing throughout the paper, and these examples are abstract at times. I suggest telling the story with a few recurring concrete examples, which also appear in the analyses of the paper. One such example would be mean estimation in federated learning with three levels of privacy. Another such example could be the setup with different users that have different privacy sensitivities, etc.

- The paper lacks a discussion on how to explicitly solve Eq (2) (or  Eq (22)) for the different examples that are proposed. At the very least, I suggest that (once the paper follows a few concrete examples) the authors discuss how to solve the problem for each setting. Also, please give some insight on how solving the problem might be easy/challenging for various types of utility functions, etc.

- In Eq. (15) what does a_i encode? Please develop some intuition here by solving the problem for various a_i’s and exploring the role of a_i’s in the solution. In Eq. (15), please also change the notation of a_i that clashes with the previously introduced a.

- Please make it clear how the theoretical results (theorems) are related to the existing literature. Specifically, please make it clear what the novelty of each theorem is with respect to the existing literature on Shapley values. In particular, if the theorem resembles a specific result, it would be best for the reader to be given this information explicitly right after the theorem statement.

- Please share some intuition for the specific type of utility function in (15). Is there an operational meaning for this logarithmic form? Does this have an implication in solving the problem in Eq (2)?

- As it stands, Section 4 ends abruptly. Please further develop the mean estimation example and explore the role of different parameters of the problem so that it is better understood what may incentivize different users to choose the three privacy levels. Please also further develop this example to understand the effect of number of users, the effect of task similarity, the effect of the number of datapoints each user has (which could be captured through the estimation variance of each user), etc. Currently, the choice of users for different privacy levels seems arbitrary. I was hoping to see how the “sensitivity” of the users to privacy leads them to choose different privacy levels. I was hoping to also see how the behavior of other users might affect the behavior of a given user to choose a different privacy level. As it stands, the example falls short to explain the game theoretic aspect of the developments of this paper. It would be best to relate the content of Section 5 back to the examples in Section 4.

- I was curious whether the mean estimation in FL was related to the simple theoretical setting that has been extensively used and analyzed in the FL literature, e.g., in [1-3]. I wonder if you may be able to adapt the privacy framework by incorporating varying levels of differential privacy as distinct sharing tiers in one of your examples.

[1] Li, Tian, et al. "Ditto: Fair and robust federated learning through personalization." International Conference on Machine Learning. PMLR, 2021.

[2] Cho, Yae Jee, et al. "Personalized federated learning for heterogeneous clients with clustered knowledge transfer." arXiv preprint arXiv:2109.08119 (2021).

[3] Aldaghri, Nasser, et al. "Federated learning with heterogeneous differential privacy." arXiv preprint arXiv:2110.15252 (2021).

- Right before (8), there is a reference to Eq. (33). Please make sure that the paper is self-contained for reading purposes.

- Please explain the axioms in words and make sure the reader intuitively understands what they entail and why they are sensible. I suggest giving the axioms one by one followed by a justification for each before moving to the next one.

- The statement regarding computational complexity at the beginning of page 8 seems out of place. Please expand on why that is important and how it is related.

- I was hoping to see an example where the platform is incentivized to pay different users differently because of the way the platform estimates the perceived value of each user’s data. Does such a setting fall within the realm of the proposed framework? If so, please capture it within (a modification to) one of the proposed examples?

- The example in Section 5.1 is quite nice. Please expand the analysis here to further understand the impact of many other parameters of the problem on the design of the mechanism to develop intuition for the reader.

- In the real world, the users are evolving. The privacy sensitivity of an existing user may change over time. Users with other privacy sensitivities might join the system over time, etc. It would be great if the paper can shed light on how to understand evolution of the incentives in such settings. To make the problem more manageable, let’s consider two settings: (a) One user changes their privacy sensitivity, and understand how the solution changes. (b) A new user is introduced into the system and analyze how the mechanism works given the sensitivity level of this user.

- Please clarify the differences between privacy level 2 and 3 first mentioned on Page 1, for example, what types of data statistics the server can query for, in what scenarios the query results will (and will not) allow the server to reconstruct raw data, and how many times the server can make such queries. Please define the exact privacy model somewhere in the paper. It mentions different levels of privacy conceptually throughout the paper, and also defines DP in Definition2; and it is not clear how we should reason about the actual privacy definition (e.g., how does DP fit into part of the story). It is also not clear if level 2 (rho=1) is defined with secure aggregation built in or not. If yes, then need to justify it; if sometimes, then need to define when.

- Please consider adding a table of notations to make it easy for the reader to find the meaning of different parameters.

**Audience:**

The paper is of interest to the TMLR audience, in part as evidenced by the engaging reviews of 4 existing reviewers from the pool.

**Claims And Evidence:**

The claims and evidence could still be improved. Please see the detailed comments below.

---

> ### Author Response · Authors · 2023-12-21
> **Extension Request**
>
> Dear AE and Reviewers,
>
> We believe we can address these concerns in our revision. Due to the overlap with the end of term, conferences and holidays, we would like to request a two week extension.
>
> Regards,
>
> Paper 1415 Authors

---

> > ### Comment · Action_Editors · 2023-12-21
> > **Extension granted**
> >
> > Dear authors,
> >
> > Given the circumstances, we have now extended the deadline for the submission of the revision to Jan 15. We hope that works, and please let us know in the mean time if you have any other questions.
> >
> > Best,\
> > Ahmad